# Calcium oscillations coordinate feather mesenchymal cell movement by SHH dependent modulation of gap junction networks

Ang Li[1,2], Jung-Hwa Cho[3,4], Brian Reid[5], Chun-Chih Tseng[1,6], Lian He[7], Peng Tan[7], Chao-Yuan Yeh[1,8], Ping Wu[1], Yuwei Li[9], Randall B. Widelitz[1], Yubin Zhou ⓘ [7], Min Zhao ⓘ [5], Robert H. Chow[3] & Cheng-Ming Chuong[1,8]

Collective cell migration mediates multiple tissue morphogenesis processes. Yet how multi-dimensional mesenchymal cell movements are coordinated remains mostly unknown. Here we report that coordinated mesenchymal cell migration during chicken feather elongation is accompanied by dynamic changes of bioelectric currents. Transcriptome profiling and functional assays implicate contributions from functional voltage-gated $Ca^{2+}$ channels (VGCCs), Connexin-43 based gap junctions, and $Ca^{2+}$ release activated $Ca^{2+}$ (CRAC) channels. 4-Dimensional $Ca^{2+}$ imaging reveals that the Sonic hedgehog-responsive mesenchymal cells display synchronized $Ca^{2+}$ oscillations, which expand progressively in area during feather elongation. Inhibiting VGCCs, gap junctions, or Sonic hedgehog signaling alters the mesenchymal $Ca^{2+}$ landscape, cell movement patterns and feather bud elongation. $Ca^{2+}$ oscillations induced by cyclic activation of opto-cCRAC channels enhance feather bud elongation. Functional disruption experiments and promoter analysis implicate synergistic Hedgehog and WNT/β-Catenin signaling in activating *Connexin-43* expression, establishing gap junction networks synchronizing the $Ca^{2+}$ profile among cells, thereby coordinating cell movement patterns.

[1] Dept. of Pathology, USC Keck School of Medicine, Los Angeles, CA 90033, USA. [2] Dept. of Kinesiology, University of Texas at Arlington, Arlington, TX 76019, USA. [3] Dept. of Physiology & Biophysics, USC Keck School of Medicine, Los Angeles, CA 90033, USA. [4] National Institute of Neurological Disorders and Stroke, Bethesda, Maryland 20824, USA. [5] Dept. of Dermatology, UC Davis School of Medicine, Sacramento, CA 95817, USA. [6] Dept. of Biochemistry & Molecular Biology, USC Keck School of Medicine, Los Angeles, CA 90033, USA. [7] Institute of Biosciences and Technology, College of Medicine, Texas A&M University, Houston, TX 77030, USA. [8] Integrative Stem Cell Center, China Medical University Hospital, 404 Taichung, Taiwan. [9] Division of Biology and Biological Engineering, California Institute of Technology, Pasadena, CA 91125, USA. These authors contributed equally: Ang Li, Jung-Hwa Cho. Correspondence and requests for materials should be addressed to R.H.C. (email: rchow@med.usc.edu) or to C.-M.C. (email: cmchuong@med.usc.edu)

Collective cell migrations play key roles in gastrulation, organogenesis, wound healing, and immune responses, as well as pathological processes including chronic inflammation and cancer invasion[1,2]. Tissues undergo various types of collective cell migration. Epithelial cells, for example, have been observed to migrate in lines, sheets, strands, and hollow tubes. They rely on stable cell-cell junctions (especially adherens junctions) to maintain cooperativity. In contrast, migratory mesenchymal cells only have transient cell-cell contacts[1,2]. This could be problematic when the cell density is high or the migration distance is long (millimeter or centimeter range). Either situation greatly limits guidance cues available to cells at the rear of the migrating cohort[2,3]. Therefore, biological systems must have developed mechanisms to boost and relay directional signals.

Externally applied electric fields were found to guide directional migration of cultured cells[4]. Recent studies revealed that long-range, self-sustained $K^+$ oscillations coordinate collective proliferation and migration in bacteria[5,6]. Endogenous direct-current (DC) electric fields (EFs) have also been detected during embryogenesis/regeneration in eukaryotes and these EFs were implicated in instructing cells with directional or positional information[7–10]. However, the molecular understanding of these phenomena is rudimentary, mainly due to a lack of tools to monitor endogenous electric fields with high spatiotemporal resolution in vivo. The development of tools such as vibrating probes[11] and genetically encoded voltage- and $Ca^{2+}$ sensors[12,13] enables in-depth investigation of bioelectric signals in vivo.

Feather bud elongation in chicken dorsal skin explants is a very robust and precise biological process, even without the embryonic microenvironment[14,15]. The robustness of this process implies the maintenance of localized and stringent molecular mechanisms for

coordinating collective cell behaviors. Previous studies of cellular events implicate polarized mesenchymal cell rearrangements in directed feather bud elongation[16]. Sparse BrdU- or TUNEL-positive cells in feather mesenchyme imply minimal involvement of cell divisions or apoptosis, a potentially confounding interpretation[16,17]. We hypothesize that tissue endogenous bioelectric signals mediated by ion channels, exchangers and pumps may carry out the signal relay function in mesenchymal cells during feather elongation.

In this study we observed dynamic changes of bioelectric currents in developing chicken embryos. Before feather bud elongation, EF endogenous to dorsal skin was relatively homogenous and exhibited inward directionality. At the onset of elongation, outward electric current emerged at the anterior side of each feather bud, implying a heterogenization of the EF into multiple smaller electric circuits. Tissue-wide long-range $Ca^{2+}$ oscillations were observed in bud mesenchyme. Dampening these oscillations or introduction of exogenous oscillations altered feather morphology. Feather mesenchymal cell movement changes direction markedly when voltage-gated $Ca^{2+}$ channels (VGCCs) or gap junctions were inhibited. The landscape of the Connexin-43 based gap junction network was modulated by synergistic actions of SHH and WNT signaling. This network electrically coupled mesenchymal cells expressing heterogeneous levels of VGCCs and CRAC channels, thereby allowing synchronized $Ca^{2+}$ oscillations to occur and coordinate directional cell movements.

## Results

**Electric currents and ion channel expression in feathers.** A vibrating probe[11] was used to measure endogenous DC electric currents on intact chicken embryos from Hamburger & Hamilton[18] (H&H) stage 29–38 (Fig. 1a). Measurement locations included interbud regions (between feather buds), anterior and

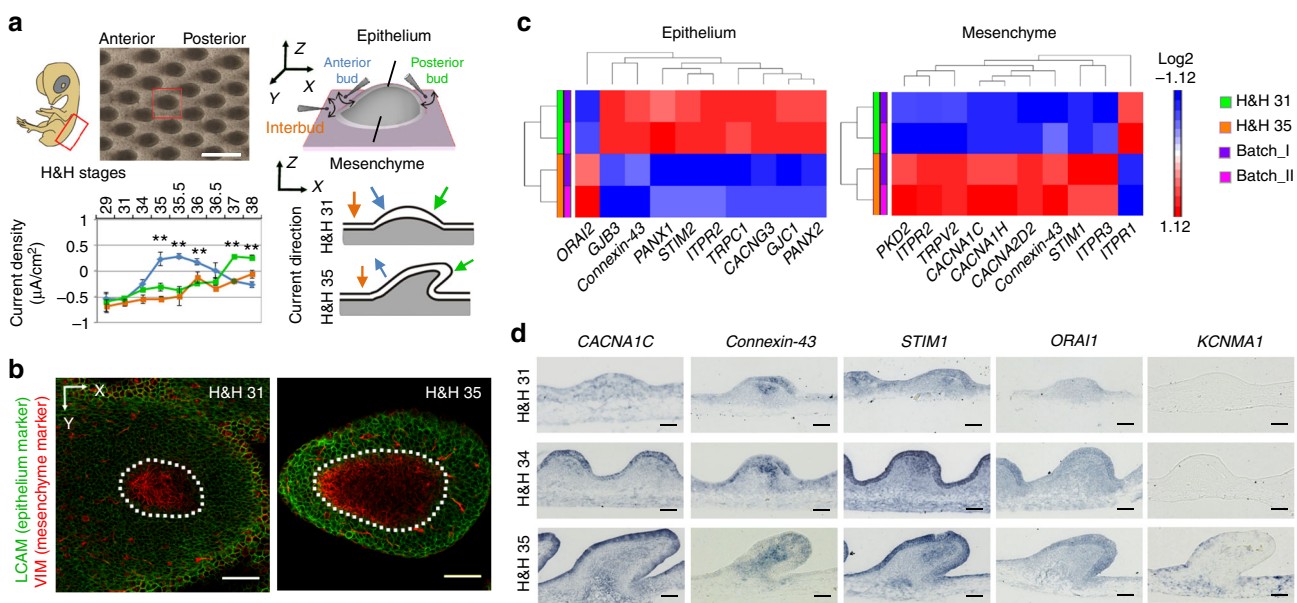

**Fig. 1** Dynamic bioelectric currents and expression of $Ca^{2+}$-related channels in elongating feathers. **a** Endogenous electric current density in anterior, posterior feather buds and interbud regions in intact chicken embryos at different developmental stages ($n = 9$–11 measurements per position per stage). Between Hamburger & Hamilton (H&H) Stage 34–35, the direction of currents at the anterior buds reversed from inward to outward, while that at the posterior buds stayed inward. The direction of currents at interbud regions stayed inward throughout the developmental stages examined. Data are presented as mean ± s.e.m. **$P < 0.005$ (two-sample Student's $t$ test). Scale bar, 500 μm. **b** Confocal image of H&H 31 and H&H 35 feather buds stained with LCAM and VIM to reveal distinct morphology of epithelial and mesenchymal cells. VIM positive cells in epithelium are melanocytes and periderm cells. Dashed lines highlight epithelial-mesenchymal boundaries. Scale bars, 50 μm. **c** Screening of genes encoding ion channels in RNA-Seq highlights multiple $Ca^{2+}$-conducting channels. **d** In situ hybridization results of genes encoding components of VGCCs, gap junctions, CRAC channels, and $Ca^{2+}$-activated $K^+$ channels. Scale bar, 50 μm

posterior feather bud regions. Inward currents were observed in interbud regions for all developmental stages examined. Inward ionic currents were also observed in feather buds before the initiation of polarized elongation (H&H 29–34). Following the onset of elongation (H&H 34–35), the current direction at the anterior bud regions reversed, while current directionality remained inward at posterior bud regions. We hypothesize that these dynamic changes of endogenous feather bud ionic currents result from spatiotemporally regulated ion channel expression and activity.

To screen candidates for the ion channels involved in stage-dependent current changes, we compared the transcriptomes of H&H 31 and H&H 35 embryonic chicken dorsal skins (Supplementary Fig. 1), with epithelium and mesenchyme separated due to their distinct cell morphology and molecular expression (Fig. 1b). Epithelium is primarily comprised of cuboidal cells tightly arranged in a honeycomb pattern (highlighted by staining of LCAM, Liver Cell Adhesion Molecule), while mesenchyme is mainly composed of VIM (Vimentin) positive, bipolar, or multipolar fibroblasts (Supplementary Fig. 2a, b). Neurites are very rare and only exist in mesenchyme underneath feather buds during feather elongation (Supplementary Fig. 2c). We focus on $Ca^{2+}$ channel coding genes in RNA-Seq analysis, because $Ca^{2+}$ ions not only serve as carriers of inward currents[19], but also second messengers to link membrane potentials to downstream cellular events[20]. Intercellular gap junction channels were also included, as they are crucial to establish tissue-wide electrical paths[21]. $Ca^{2+}$-activated $K^+$ channels carrying outward currents may help restore the resting membrane potential after depolarization-induced $Ca^{2+}$ influx[22].

To shorten the candidate gene list (Supplementary Data 1), the following filter condition was applied: minimum RPKM (reads per kilobase of transcript per million mapped reads) > 1.5, fold change > 1.3 (before Log2 transformation), $P < 0.05$ (Fig. 1c). Afterwards, RNA in situ hybridization and RT-qPCR were performed to validate the RNA-Seq results (Fig. 1d and Supplementary Fig. 3). L-type and T-type voltage-gated $Ca^{2+}$ channels (VGCCs), gap junctions and $Ca^{2+}$ release activated $Ca^{2+}$ (CRAC) channels exhibited developmental stage-dependent variations of expression. Connexin-43 gap junctions were localized at the posterior–distal epithelium and mesenchyme until feather elongation when the expression region expanded in the mesenchyme. Meanwhile their expression level decreased in the epithelium. STIM1 (Stromal Interaction Molecule 1, $Ca^{2+}$ sensor of CRAC channels) was expressed with a similar pattern as Connexin-43 in feather mesenchyme. The binding partner of STIM1 (ORAI1, ion conducting pore subunit of CRAC channels) was also present in feathers but exhibited no stage-dependent variation in expression. Furthermore, in situ hybridization detected sparse KCNMA1 (component of $Ca^{2+}$-activated $K^+$ channel) expression in anterior epithelium and basal mesenchyme of H&H 35 feather buds.

**Functional VGCCs and Connexin-43 gap junctions in feathers.** To visualize cytoplasmic $Ca^{2+}$ buildup due to influx through functional VGCCs in feathers, we constructed an avian viral vector (RCAS) expressing $Ca^{2+}$ sensor GCaMP6s[10], T2A peptide, and mCherry. By normalizing the fluorescence intensity of GCaMP6s to that of mCherry, changes in $Ca^{2+}$ levels could be measured and compared among tissues despite heterogeneous levels of viral expression. To visualize cytosolic $Ca^{2+}$ changes along both the transverse and sagittal planes, we adopted skin explants and strip configurations (Fig. 2a). As expected, KCl application to depolarize membrane potentials triggered $Ca^{2+}$ increases in mesenchymal but not epithelial cells. The responses

elevated as the feathers developed (Fig. 2b–f, k; Supplementary Fig. 4a; Supplementary Movies 1–5). Pretreating skins with a VGCC blocker, Nifedipine, dramatically reduced the KCl response (Fig. 2g, k and Supplementary Movie 6). To measure direct currents through VGCCs, H&H 35 mesenchyme was dissociated to single cells for patch clamp experiments (Supplementary Fig. 5). Consistent with the in situ hybridization data (Supplementary Fig. 3a), a current–voltage (I–V) plot supports the presence of functional VGCCs. The measured I–V curve indicates that the cells mainly expressed T-type VGCCs, but we cannot exclude the possibility that a cell population may express L-type VGCCs due to the limited number of tested cells used in the patch clamp experiment. KCl-induced fast $Ca^{2+}$ increases were also observed in cultured single cells (Fig. 3a, b ROI 1 and Supplementary Movie 7). Taken together, our data suggest that active VGCCs are present in a subpopulation of feather mesenchymal cells.

To confirm the presence of functional gap junction channels, we performed the scrape-loading/dye transfer assay to confirm the permeability of gap junctions in feathers[23]. In H&H 35 feather buds Lucifer yellow quickly spread to almost all feather mesenchymal cells after scraping (Supplementary Fig. 6a, b). Pretreatment of the skin for 1 h with 18-α-GA, which disrupts the Connexin-43 arrangement within the channel[24], dramatically reduced intercellular transfer of Lucifer yellow (Supplementary Fig. 6a). Similar results were observed in samples treated with PMA (Supplementary Fig. 6b), which activates PKC to alter Connexin-43 phosphorylation and downregulates Connexin-43 expression[25–28]. As gap junction communication plays a role in propagating electrical signals among cell populations[29], we expected weaker KCl responses in feathers after administering gap junction blockers. Indeed, pretreating the skins with either Carbenoxolone (an 18-α-GA derivative) or PMA significantly reduced the KCl responses ($P < 0.05$, Wilcoxon rank test, Fig. 2h, k, Supplementary Fig. 6c and Supplementary Movies 8, 9). These data indicate that a functional gap junction network exists in feather mesenchyme.

**Functional CRAC channels in elongating feathers.** CRAC channels are activated by depleting the endoplasmic reticulum (ER) $Ca^{2+}$ stores. To assess CRAC channel expression in feathers, intact skin explants were bathed in EGTA-buffered $Ca^{2+}$ free solution containing Thapsigargin (ER $Ca^{2+}$ pump blocker). After Thapsigargin treatment, 2 mM $Ca^{2+}$ Ringer's solution was applied to induce store-operated $Ca^{2+}$ influx. Notable $Ca^{2+}$ influx was observed in both feather epithelium and mesenchyme (Fig. 2i, k and Supplementary Movie 10). The CRAC channel inhibitor, BTP2[30], dramatically reduced $Ca^{2+}$ influx (Fig. 2j, k and Supplementary Movie 11). Taken together, feather buds express functional CRAC channels.

In dissociated single mesenchymal cells, uncoordinated, spontaneous, and long-lasting $Ca^{2+}$ fluctuations were observed (Fig. 3a ROI 2, 3). Normally, high KCl stimulation causes a rapid rise in $Ca^{2+}$, owing to membrane depolarization leading to opening of VGCCs. This was, indeed, observed in many cells (Fig. 3a, b ROI 1). Surprisingly, KCl-induced $Ca^{2+}$ decreases were also observed in some cultured mesenchymal cells (Fig. 3a, b ROI 3). We hypothesize that the decrease might be due to membrane depolarization shutting off $Ca^{2+}$ influx through the inwardly rectifying CRAC channels[31]. Some cells exhibited biphasic responses to KCl (a rapid rise, followed by a decrease), possibly due to the cells' expressing both VGCCs and CRAC channels (Fig. 3a, b ROI 2). When KCl was applied along with Nitrendipine, a VGCC blocker, $Ca^{2+}$ levels showed only the decrease, confirming the role of VGCCs in the rapid upward

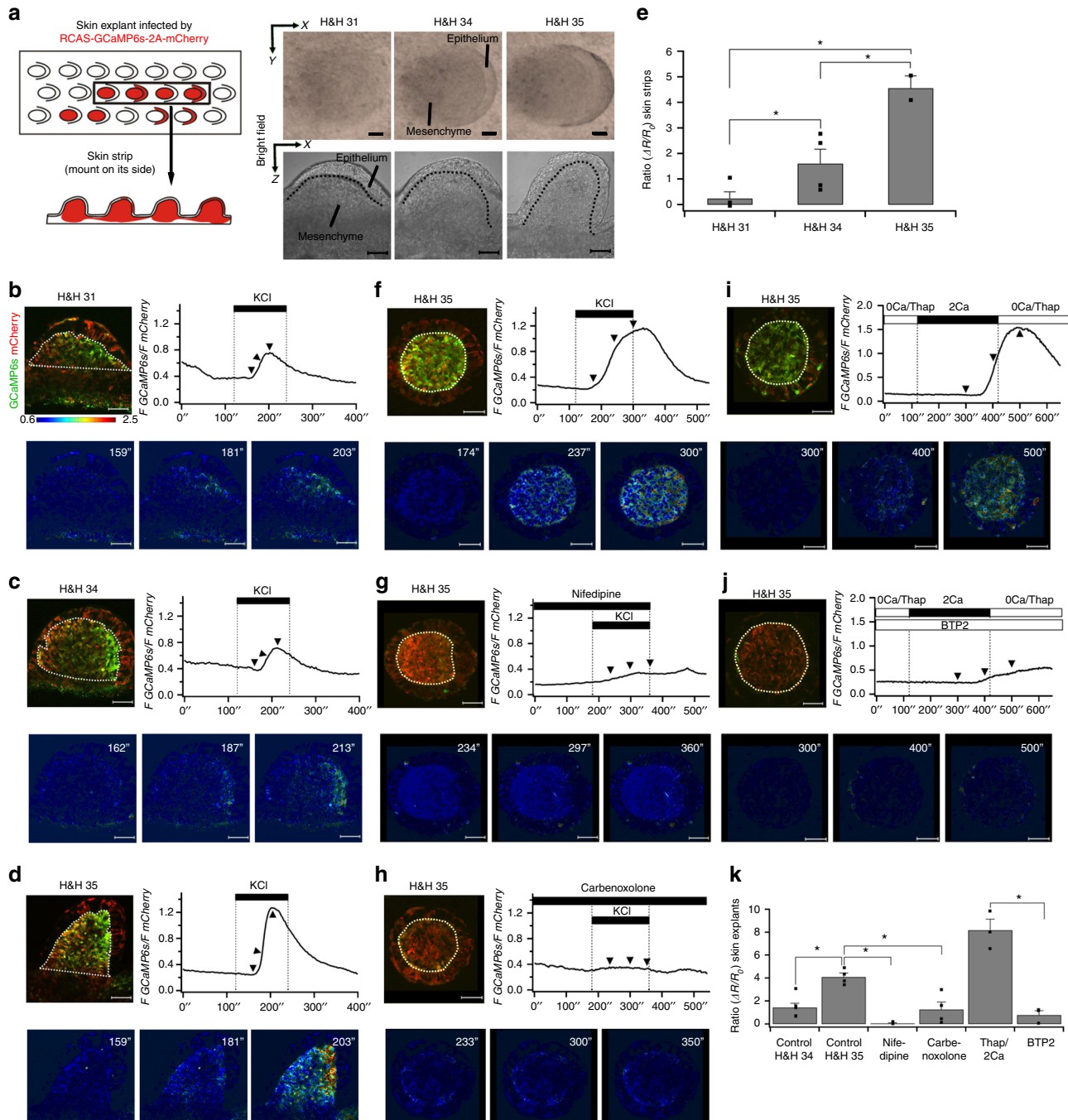

**Fig. 2** Functional VGCCs, gap junctions and CRAC channels in elongating feathers. **a** Schematic drawing exhibiting the skin explant and strip configuration used for time-lapse $Ca^{2+}$ imaging. Bright field images of feather buds at different developmental stages in skin explants and strips are shown. Dotted lines highlight epithelial-mesenchymal boundaries. Scale bars, 50 μm. **b–d** Time-lapse $Ca^{2+}$ imaging of feathers from skin strips at different stages ($n = 3$ for each stage). Dotted lines indicate regions of interest (ROI) for quantification ('' denotes seconds). Pseudocolor images of the ratios are presented at selected time points (arrowheads). Scale bar, 50 μm. **e** Quantifying change of mesenchymal $Ca^{2+}$ increases in feathers from skin strips. $R_o$: baseline $F$ GCaMP6s/$F$ mCherry ratio before KCl application. $\Delta R$: the peak ratio subtracts $R_o$. Mean ± s.e.m. *$P < 0.05$ (Wilcoxon rank test). Dots represent individual data points. **f** Time-lapse $Ca^{2+}$ imaging of feathers in skin explants at H&H 35 ($n = 16/16$). **g** Pretreatment with 50 μM Nifedipine for 3 min greatly dampened the KCl-induced $Ca^{2+}$ elevation ($n = 4/4$). **h** Pretreatment with 150 μM Carbenoxolone for 30 min significantly reduced the KCl response ($n = 4/4$). **i** Feather buds pretreated for 30 min with 0 mM $Ca^{2+}$ solution including 10 μM Thapsigargin exhibited notable $Ca^{2+}$ influx after perfusion of 2 mM $Ca^{2+}$ solution ($n = 3/3$). **j** 5 μM BTP2 significantly reduced the $Ca^{2+}$ influx ($n = 3/3$). **k** Quantification of $Ca^{2+}$ response in feathers from skin explants under different treatments. Mean ± s.e.m. *$P < 0.05$ (Wilcoxon rank test)

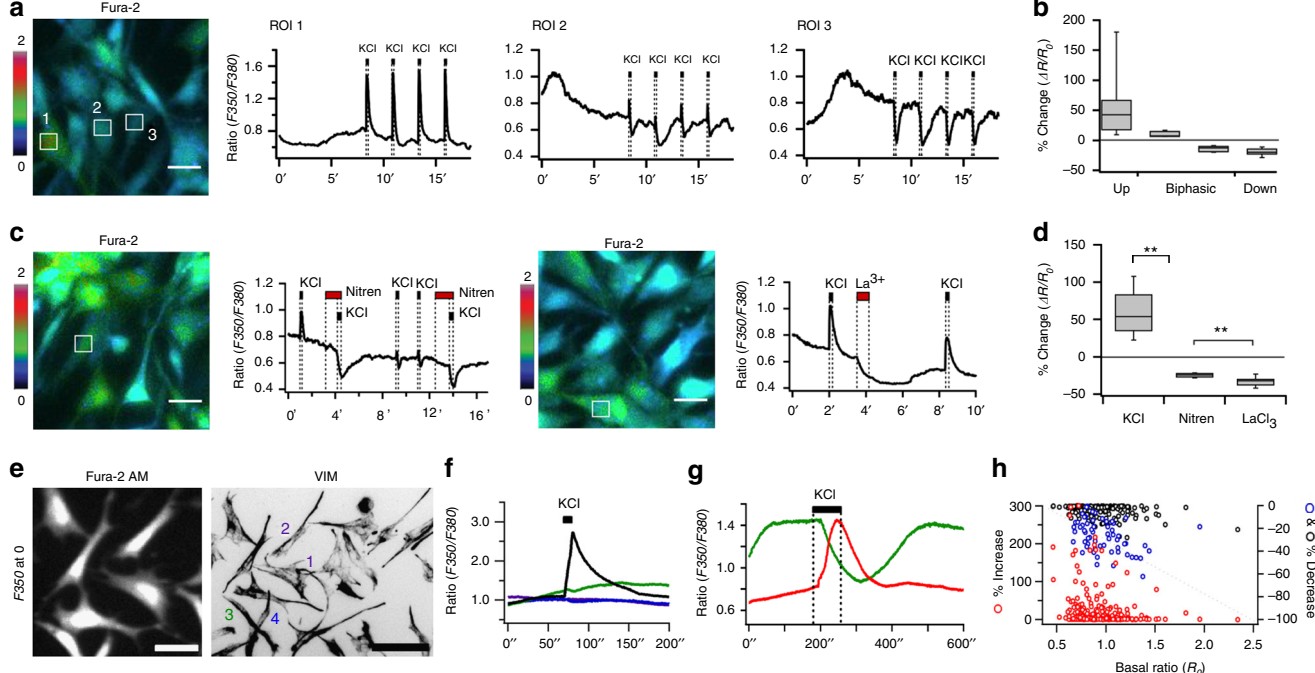

**Fig. 3** Heterogeneous Ca$^{2+}$ responses in cultured mesenchymal cells. **a, b** Three types of Ca$^{2+}$ responses upon 100 mM KCl stimuli in H&H 35 single mesenchymal cells: KCl-induced Ca$^{2+}$ increase (ROI 1, $n = 24/73$), biphasic Ca$^{2+}$ response (ROI 2, $n = 7/73$), KCl-induced Ca$^{2+}$ decrease (ROI 3, $n = 42/73$). Rectangles denote regions of interest (ROI) for quantification. AM-Fura-2 dyes were used to measure Ca$^{2+}$ level by excitation wavelength of 350 nm ($F350$) and $F380$. Scale bar, 25 μm. ' denotes minutes. Whisker bottom, box bottom, median line, box top, and whisker top correspond to 10, 25, 50, 75, and 90 percentile. **c, d** 10 μM Nitrendipine (Nitren) or 10 μM LaCl$_3$ was used to inhibit L-type VGCC ($n = 5$) and CRAC channels ($n = 20$), respectively. **$P < 0.01$ (Wilcoxon rank test). **e** Mesenchymal cells were identified by vimentin staining after Ca$^{2+}$ measurements. Scale bars, 35 μm, 110 μm. **f** Fast application of KCl solution for 10 s induced immediate Ca$^{2+}$ responses. Four cells labeled with colored numbers in **e** were chosen to plot the ratios. '' denotes seconds. **g** Slow, gentle bath perfusion of KCl (indicated by the time delay between onset of KCl application and response) implies the response is not an artifact of shear stress stimulation. **h** Negative correlation between $R_o$ and KCl-induced Ca$^{2+}$ increase (Red circle, fast application/bath perfusion, $n = 246$). Positive correlation between $R_o$ and KCl-induced Ca$^{2+}$ decrease (Black circle, fast application, $n = 184$; blue circle, bath perfusion, $n = 62$). % increase or decrease of Ca$^{2+}$ response was calculated by $\Delta R/R_o$

spike in the biphasic response ($P < 0.01$, Wilcoxon rank test, Fig. 3c, d). The KCl-induced decrease, even with Nitrendipine, suggests that the cells may express constantly active CRAC channels in addition to VGCCs (Fig. 3c and Supplementary Movie 12). Application of lanthanum (La$^{3+}$), a CRAC channel pore blocker, significantly decreased the baseline cytoplasmic Ca$^{2+}$ levels in some cells, which also indicates the presence of sustained active CRAC channel currents ($P < 0.01$, Wilcoxon rank test, Fig. 3c, d and Supplementary Movie 13). Additionally, heterogeneous responses to Ca$^{2+}$ stimulation after Ca$^{2+}$ deprivation were observed in the dissociated mesenchymal cells, confirming the non-uniform presence of CRAC channels (Supplementary Fig. 7a-d). Finally, biphasic KCl responses were also observed in feather bud mesenchyme under skin strip configuration (Supplementary Fig. 8a, b and Supplementary Movie 14), implying the presence of active CRAC channels in vivo.

While we tried to classify the subpopulations of the mesenchymal cells with respect to KCl-induced Ca$^{2+}$ responses, we discovered that there was a correlation between cell morphology and Ca$^{2+}$ responses. Surface area and aspect ratio measured in VIM staining images indicated that smaller- and/or thin-shaped cells tended to exhibit elevated baseline Ca$^{2+}$ levels and KCl-induced Ca$^{2+}$ decreases, while larger- and/or polygonal-shaped cells tended to exhibit KCl-induced Ca$^{2+}$ increases (Fig. 3e–h and Supplementary Fig. 7e, f). The elevation of baseline Ca$^{2+}$ could be abolished by Thapsigargin treatment,

implying contributions from constantly active CRAC channels (R$_0$ of Fura-2 is about 0.5 in all cells measured in Supplementary Fig. 7a, b) Notably, some mesenchymal cells located at the anterior region of buds were elongated compared to ones located at the posterior region (Supplementary Fig. 2a, b). Therefore, we speculated that subpopulations of dissociated mesenchymal cells somehow retain their endogenous locational information which causes them to express the functional Ca$^{2+}$ channels corresponding to their sites of origin.

To further evaluate the contribution of CRAC channels to the dynamic Ca$^{2+}$ changes in feather buds, we created light-activated CRAC channels called opto-cCRAC by fusing the chicken cytosolic STIM1 domain with the LOV domain from *Avena sativa* phototropin 1[32]. STIM1 is located in the ER membrane, where normally, upon depletion of the ER, it activates the plasma membrane ORAI channels. Upon blue light illumination, the LOV domain undergoes conformational changes, exposing the STIM1 cytosolic domain to engage ORAI channels on the cell membrane. We confirmed that the opto-cCRAC can induce Ca$^{2+}$ influx in Hela cells (Fig. 4a and Supplementary Movie 15) and H&H 35 mesenchymal cells (Fig. 4b and Supplementary Movie 16). When we artificially elicited Ca$^{2+}$ oscillations by cyclic activation of opto-cCRAC in skin explants transduced with RCAS virus encoding opto-cCRAC for 48 h, the highly infected feathers were significantly elongated compared to the poorly infected ones on the contralateral side ($P < 0.01$, Wilcoxon rank test, Fig. 4c). Control skins transduced with RCAS virus encoding

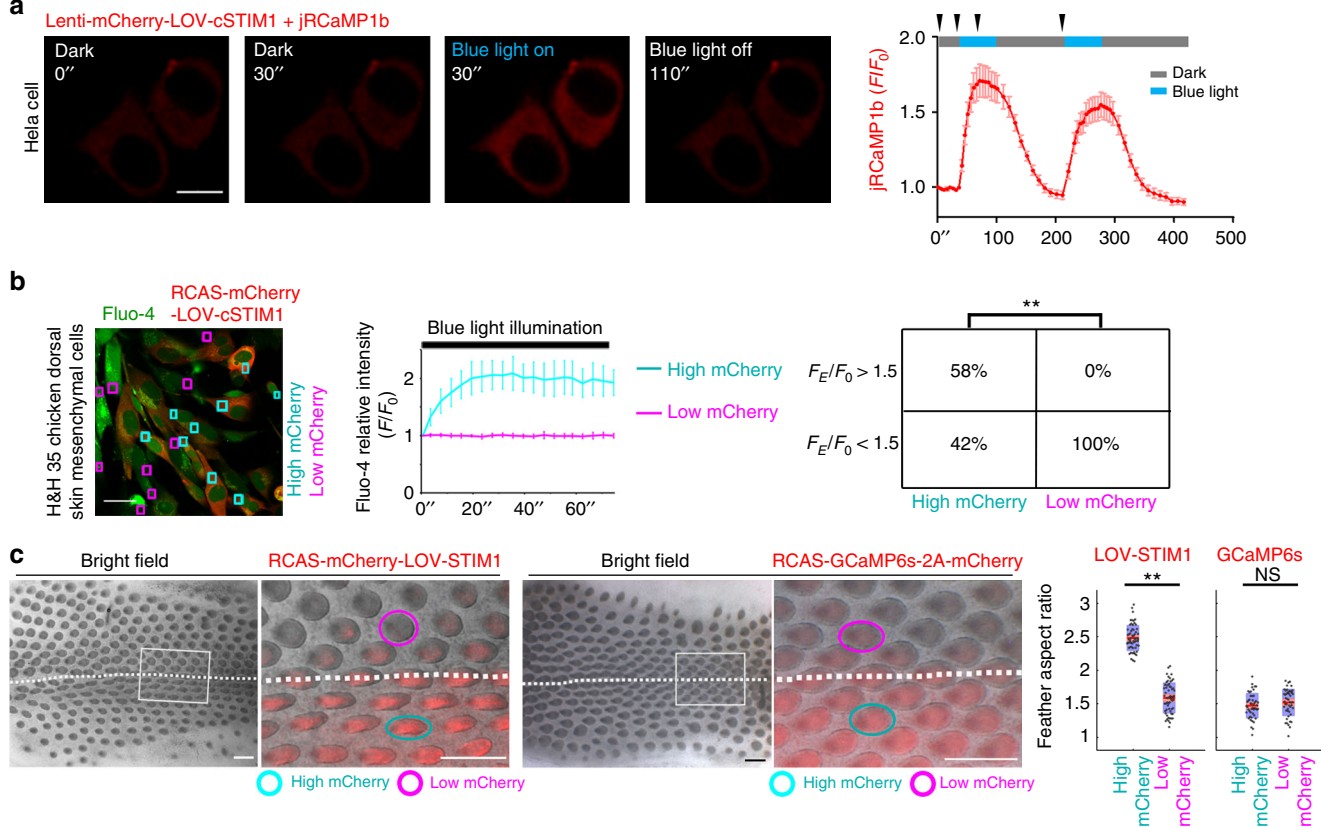

**Fig. 4** Ca²⁺ influx induced by cyclic activation of opto-cCRAC enhances feather elongation. **a** In Hela cells opt-cCRAC induced Ca²⁺ influx after blue light exposure ($n = 20/20$). jRCaMP1b was expressed to detect Ca²⁺ level changes. Arrowheads in the line plot highlight the time point of images on the left. $F/F_0$, changes in fluorescence. Mean ± s.d. Scale bar, 10 μm. **b** Light-induced Ca²⁺ influx occurred in H&H 35 mesenchymal cells infected by virus encoding opto-cCRAC (cyan rectangles, $n = 25/43$) but not in the uninfected cells (magenta rectangles, $n = 0/33$). Mean ± s.d. Scale bar, 10 μm. ('' denotes seconds). $F_E$: fluorescence at the end of recording (70''); **$P < 0.01$ (Chi-square test). **c** After 48 h cyclic blue light illumination, skin regions ($n = 8/10$) highly expressing opto-cCRAC developed more elongated feathers than the poorly infected buds ($n = 60$ for each group, **$P < 0.01$, Wilcoxon rank test). No discernable feather morphology changes ($n = 48$, NS, not significant, Wilcoxon rank test) were observed in skins infected by RCAS-GCaMP6s-T2A-mCherry. Regions close to the dorsal mid-line were magnified (rectangles) to compare feather aspect ratio (ellipses). Scale bar, 500 μm. Customized boxplot: Mean (red) ± s.d. (pink), 95% confidence interval (violet). Dots denote individual data points

GCaMP6s-T2A-mCherry exhibited no discernible feather morphology changes. Thus collective cell Ca²⁺ oscillations are crucial for feather elongation.

**Ca²⁺ oscillation patterns modulate feather elongation.** To directly visualize the dynamics of the tissue Ca²⁺ profile, we did 4D (3D space + time) imaging of elongating feathers from H&H 34 skin explants and strips infected with RCAS virus encoding GCaMP6s-T2A-mCherry. Intriguingly, not only did we see fast, sporadic Ca²⁺ transients on the scale of seconds (Supplementary Fig. 8c, e and Supplementary Movie 17), we also observed slow Ca²⁺ oscillations, which are generally synchronized (peaks and valleys in phase) in anterior and posterior mesenchyme (Fig. 5a, Supplementary Fig. 8d, e and Supplementary Movies 18–21). The oscillating zone slowly expands in the anterior–proximal direction at about 1.7 μm/min (Fig. 5a and Supplementary Fig. 8d, f). These oscillations are dramatically dampened upon the inhibition of VGCCs or gap junctions, although some sparse Ca²⁺ transients could still be spotted (Fig. 5a; Supplementary Fig. 6d and Supplementary Movies 22–27). Thus VGCCs and gap junction channels are required for multicellular Ca²⁺ oscillations during feather elongation.

Next we explored whether the alterations of physiological Ca²⁺ dynamics would change feather morphology. We treated the skin with Nifedipine, Carbenoxolone, PMA, and another gap junction inhibitor Mefloquine[33] for 4 days and compared feather

morphology to the untreated controls (Fig. 5b and Supplementary Fig. 9). The control feathers developed into elongated filaments tilting posteriorly. Nifedipine-treated feathers grew upward for a while and then stopped, without developing an apparent anterior–posterior (A–P) polarity. Carbenoxolone and Mefloquine-treated feathers were also shorter than the controls. PMA treatment induced more severe phenotypes than Carbenoxolone in that feather buds disappeared in some skin regions. This difference may result from downstream effects of PKC signaling other than gap junction inhibition. We also genetically perturbed gap junction expression levels using lentivirus encoding *Connexin-43* shRNA (pLL3.7-Cx43-SH)[34]. The infected feather buds were notably shorter than the counterparts in the contralateral, virus-negative side ($P < 0.01$, Wilcoxon rank test, Fig. 5c). All the chemicals also significantly altered the feather aspect ratio ($P < 0.01$, Wilcoxon rank test), while none of them significantly increased cell apoptosis (Supplementary Fig. 9). Nifedipine treatment increased cell proliferation while Mefloquine treatment had the opposite effect. Taken together, neither cell proliferation nor apoptosis is a major process involved in oriented feather elongation.

**VGCCs and gap junctions modulate mesenchymal cell migration.** Since feather elongation is mainly driven by cell rearrangement during development[16], we scrutinized cell movement patterns

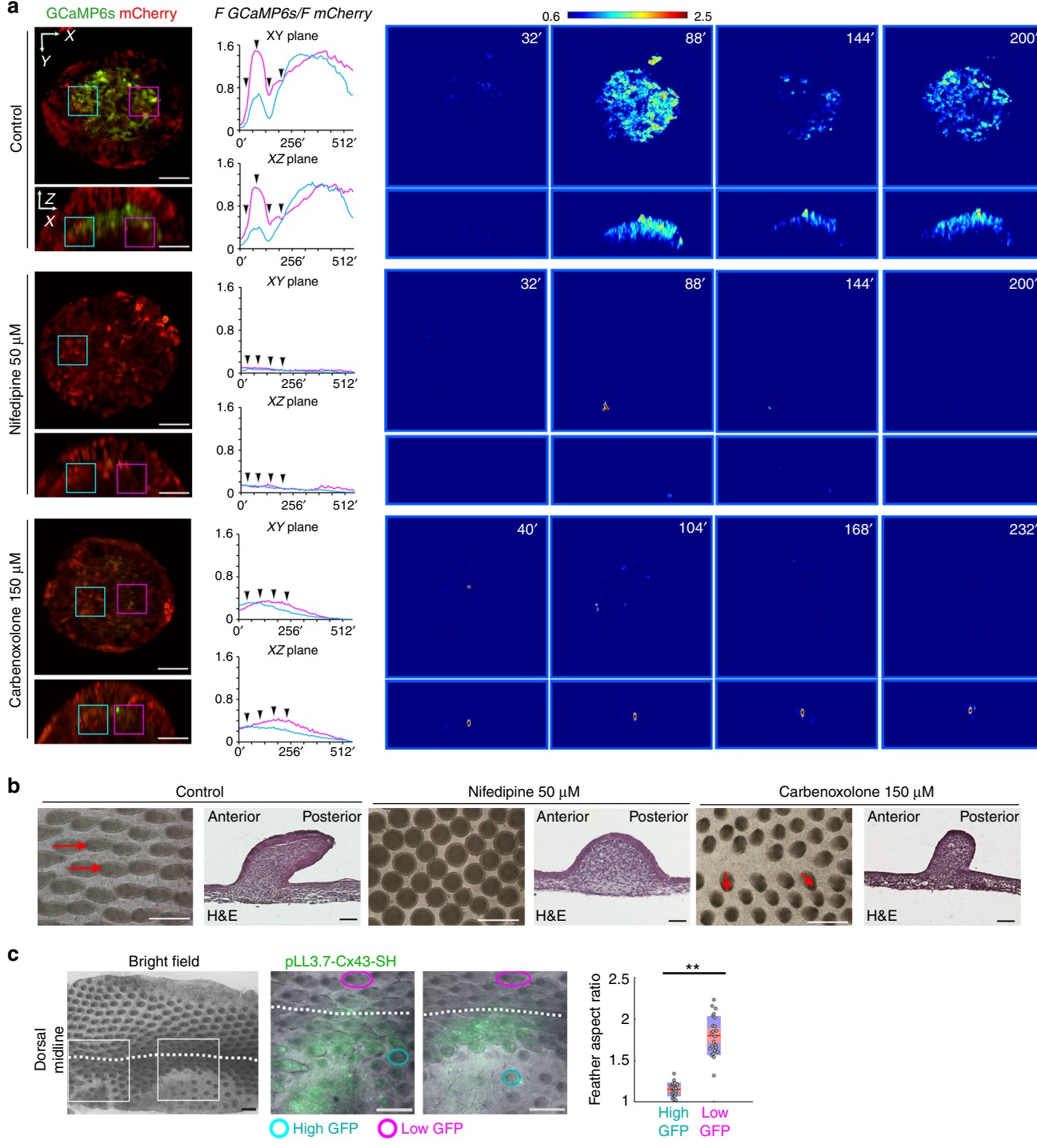

**Fig. 5** Modulating physiological Ca²⁺ oscillations alters feather elongation process. **a** Virtual sections of 4D ratiometric Ca²⁺ imaging of elongating feather buds on H&H 34 skins ($n = 3$). ROI analysis of GCaMP6s and mCherry fluorescence ratio exhibited synchronized Ca²⁺ oscillations in anterior (cyan rectangle) and posterior (magenta rectangle) mesenchyme. The high Ca²⁺ area expanded anteriorly over time (the intersection of cyan and magenta lines).′ in x-axis denotes minutes. Pseudocolor ratiometric images were shown at selected times (arrowheads). 50 μM Nifedipine or 150 μM Carbenoxolone treatment dramatically reduced the number of cells showing elevated Ca²⁺ ($n = 2$). Scale bar, 50 μm. **b** Feather buds on H&H 31 skins became elongated filaments oriented posteriorly after 4-day culture with DMSO (red arrows, $n = 10/10$). Nifedipine-treated feather buds were shorter and had no apparent anterior–posterior polarity ($n = 8/8$). 150 μM Carbenoxolone treatment inhibited feather elongation and disrupted feather polarities (red arrows, $n = 6/6$). H&E: Hematoxylin and Eosin staining. Scale bars, 500 μm (whole-mount skin), 50 μm (section). **c** In H&H 35 Skins, the regions ($n = 4/4$) infected with lentivirus encoding *Connexin-43* shRNA (pLL3.7-Cx43-SH) exhibited dramatically inhibited elongation of young feather buds (cyan ellipses) compared to the control side (magenta ellipses, $n = 23$ for each group, **$P < 0.01$, Wilcoxon rank test). Rectangles highlight magnified area. Scale bar, 500 μm. Customized boxplot: Mean (red) ± s.d. (pink), 95% confidence interval (violet). Dots denote individual data points

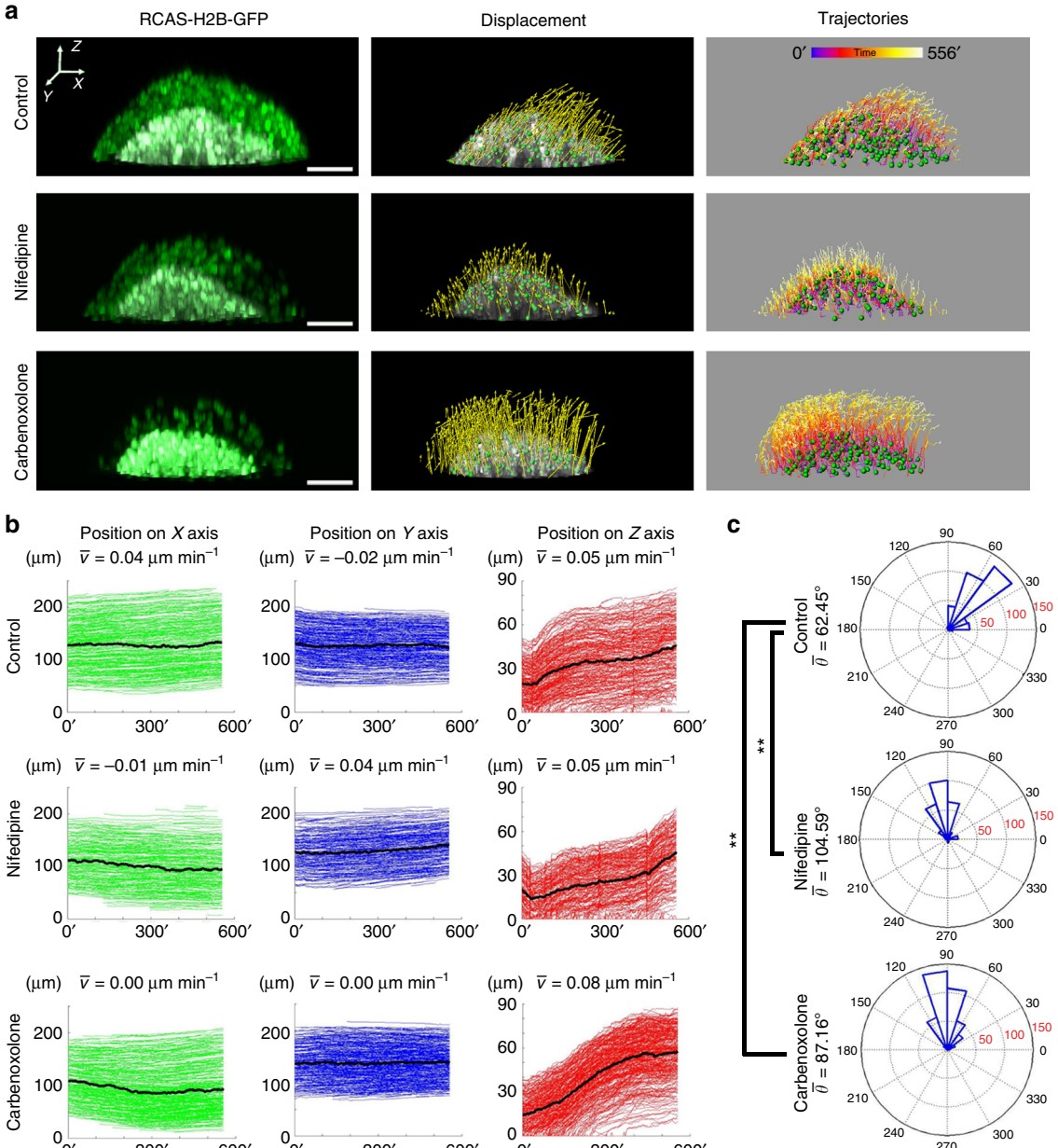

**Fig. 6** Altered feather mesenchymal cell migratory patterns upon inhibition of VGCCs and gap junctions. **a** 4D cell nucleus (H2B-GFP) imaging of elongating feather buds on H&H 34 skins. Left-most panel, 3D rendering of the feather buds at Time 0, with the mesenchyme highlighted in white. Middle panel, Mesenchymal cell nuclei were segmented (green dots) and their displacement vectors (yellow arrows) were mapped. Right panel, Cell movement trajectories. Different trajectory colors represent time (' denotes minutes). Scale bar, 50 μm. **b** Plotting individual mesenchymal cell positions along the $X$ (anterior–posterior), $Y$ (left–right), and $Z$ axis (up–down) over time. Black line indicates averaged cell position over time. The averaged velocities ($\bar{v}$ is the averaged displacement along the corresponding axis divided by time) are shown above each plot. **c** Angle histogram plots illustrate the cell displacement angle in the $XZ$ plane (0°, 90°, 180°, and 270° represent posterior, upward, anterior, and downward directions, respectively). Cells tracked: $n = 378$ for controls, $n = 316$ for Nifedipine, and $n = 278$ for PMA-treated samples. The averaged displacement angles ($\bar{\theta}$) were significantly altered upon Nifedipine or PMA treatment. **$P < 0.01$ (Watson's $U^2$ test)

by tracking individual mesenchymal cells labeled with histone-H2B GFP in skin explants. Since there was a strong correlation between $Ca^{2+}$ dynamics and feather morphology changes as shown above, we tested the effect of VGCC and Gap junction blockers on cell migration. We segmented out the feather mesenchyme to exclude epithelial cells, tracked cell positions over time, and plotted trajectories along the $X$ (anterior–posterior), $Y$ (left–right), and $Z$ (proximal–distal) axes (Fig. 6a). We also quantified the mean cell movement velocity by averaging every cell's displacement over time (Fig. 6b), as well as cell movement directions (displacement angle)

in the $XZ$ plane by calculating the inverse tangent of their displacement along the $Z$ and $X$ axes (Fig. 6c). In control skins, the feather mesenchymal cells move upward and posteriorly in a pulsatile manner, meaning there are moments when the migration halts (Fig. 6 and Supplementary Movie 28). When we compare mesenchymal cell migration patterns in the anterior and posterior bud regions, those in the anterior half move slower than their posterior counterparts, consistent with previous cell tracking data[16], but their movement directions are similar (Supplementary Table 1 and Supplementary Fig. 10a). In Nifedipine-treated skins, feather

mesenchymal cells mainly moved upward without notable A–P bias (Supplementary Movie 29). The movement velocity and displacement angle are similar between the anterior and posterior bud mesenchymal cells (Supplementary Table 1 and Supplementary Fig. 10b). Carbenoxolone treatment also diminished cell movement in the posterior direction (Fig. 6c and Supplementary Movie 30). In PMA-treated skins, feather mesenchymal cells mainly move downward and anteriorly (Supplementary Fig. 10c and Supplementary Movie 31), which explains why some feather buds disappeared in long-term culture (Supplementary Fig. 9a). Therefore, blocking VGCCs and gap junctions dramatically altered the mesenchymal cell movement patterns, especially the directionality.

**SHH signaling allows sporadic Ca²⁺ transients to synchronize.** Next, we explored the biochemical signals that modulate Ca²⁺ oscillation and cell migration patterns. The Ca²⁺ oscillations in elongating feathers initially emanate from the posterior–distal mesenchyme, which is known as the SHH signal responding zone. *SHH* is expressed in the posterior–distal epithelium while its receptor *PTCH1* is enriched in the underlying mesenchyme (Fig. 7a). Therefore, SHH protein from the feather epithelium must have diffused through the basement membrane into the mesenchyme. To confirm SHH signaling activity in the posterior–distal mesenchyme, we infected embryonic chicken skin with a virus-based reporter construct, RCAN-GBS-GFP[35]. The GLI-binding-site (GBS) drives gene expression upon SHH signal activation, while RCAN itself has no internal promoter activity. We found that GFP-positive cells co-localize with the *PTCH1*-positive region (Fig. 7a). The GFP-positive zone expanded in the anterior–proximal direction as feathers elongated, resembling the expansion pattern of synchronized Ca²⁺ oscillations. Furthermore, blocking SHH signaling with Cyclopamine[36] causes failure of feather bud polarization and elongation (Fig. 7b). This morphological abnormality could be fully rescued by adding the SHH agonist SAG[36] (Supplementary Fig. 9), and partially rescued by opto-cCRAC together with cyclic blue light illumination. The feather mesenchymal region enriched for opto-cCRAC elongated and became the new feather tips (Fig. 7c). In 4D Ca²⁺ imaging, Cyclopamine treatment inhibited synchronized Ca²⁺ oscillations, but sporadic Ca²⁺ transients were still observed (Fig. 7d and Supplementary Movies 32–34). When we track mesenchymal cells in Cyclopamine-treated feathers, they initially move upward, then stop and move slightly downward (Fig. 7e, f and Supplementary Movie 35). The anterior and posterior mesenchymal cell populations have significant differences in movement directionality: the anterior cells move more anteriorly while the posterior cells mainly move upward (*P* < 0.01, Watson's U² test, Supplementary Fig. 10d).

**SHH/WNT signaling module activates *Connexin-43* expression.** Disruption of Ca²⁺ oscillation patterns upon Cyclopamine treatment implies a role of SHH signaling in regulating either the activities or expression levels of certain Ca²⁺-related channels. Previous studies of Xenopus spinal cells implicated SHH signaling in increasing spontaneous neuronal Ca²⁺ transients[36]. We conducted similar experiments in cultured H&H 34 mesenchymal cells but observed no significant increase of spontaneous Ca²⁺ transients upon treatment with SHH N-terminus protein or SAG compared to the controls (Wilcoxon rank test, Supplementary Fig. 11a and Supplementary Movie 36). We also examined KCl response strength in feathers with short-term Cyclopamine (Supplementary Movie 37) or SAG (Supplementary Movie 38) treatment, but no notable changes were observed, either (Supplementary Figs. 4 and 11b). Therefore, we assume that SHH signaling mainly affects channel gene expression, which takes

longer to occur. We attempted to identify SHH-induced Ca²⁺ channel genes in an embryonic context through RCAS mediated *SHH* overexpression. However, this led to severe developmental defects before feather buds emerge (Supplementary Fig. 12a). Hence, we switched to cultured mesenchymal cells for the answer. qPCR results indicate 24-hr SHH N-terminus protein treatment mildly elevated *Connexin-43* and *STIM1*, but not *CACNA1C* or *PKD2* expression (Fig. 8a and Supplementary Fig. 12b). In skin explants, 48-hr SAG treatment also elevated *Connexin-43* expression in feathers, while Cyclopamine treatment had the opposite effect (Fig. 8b). The mild effect of SHH protein on *Connexin-43* expression in cultured mesenchymal cells may implicate the involvement of other epithelial factors in inducing *Connexin-43* expression, as epithelium is absent in the cell culture. Previously we discovered active WNT/β-catenin signaling in posterior feather mesenchyme[16]. During feather elongation, the WNT-responding zone (composed of nuclear β-Catenin positive cells) expands distally and partially overlaps with the SHH-responding zone (Fig. 8c). Therefore, we examined how WNT signaling regulates *Connexin-43* expression. RCAS enforced expression of constitutively active β-Catenin in chicken embryos substantially upregulated *Connexin-43* expression (Fig. 8d). Meanwhile no elevation was observed in control (RCAS-GCaMP6s-T2A-mCherry) embryos (Supplementary Fig. 12a). The RCAS-β-Catenin infected feather buds also exhibited randomized polarity as seen previously[16]. In skin explants treated with LiCl, which activates WNT/β-catenin signaling[37], we also observed elevated *Connexin-43* expression (Supplementary Fig. 12c), while the inhibition of WNT/β-catenin signaling by endo-IWR1[38] had the opposite effect (Supplementary Fig. 12c). qPCR of cultured mesenchymal cells also demonstrated an elevation of *Connexin-43* expression upon LiCl treatment, which is enhanced by SHH N-terminus protein treatment (Fig. 8a). Additionally, we searched for the SHH-responding motif (GLI-binding site) and WNT-responding motif (LEF1-binding site) in the active *Connexin-43* promoter region, which is marked by enrichment of Histone H3 Lysine 4 trimethylation[39] in ChIP-Seq analysis. We pinpointed a GLI-binding site and a LEF1-binding site in proximity (Fig. 8e), supporting the idea that SHH and WNT signaling may work together to activate *Connexin-43* transcription. Such a synergistic effect was not seen for *CACNA1C*, *STIM1*, or *PKD2* expression (Supplementary Fig. 12b).

## Discussion

Emergence of organized patterns from apparently random cells is the essence of tissue morphogenesis[40]. During tissue morphogenesis, cells navigate through a three-dimensional space to interact and organize themselves into proper configurations. Using time-lapse imaging and tracking, we observed dynamic and multi-dimensional cellular flows in the 4D context. Random and directed movements seem to co-exist in mesenchymal subpopulations. Cellular collectives form, disassemble, then reform. Cells move in one direction, stop, and then reorient. Amazingly, order gradually emerges from these seemingly chaotic movements, and organs with proper shape and orientation emerge. It is difficult to explain such complex choreography just with chemotaxis. The roles of mechanical force, bioelectric signals and ionic control in morphogenesis are now gaining increasing attention[41,42]. Here we leverage the distinct feather patterns on developing chicken skins to explore the correlation between biochemical cues, bioelectric signals and Ca²⁺ dynamics in collective cell behavior control.

Before the onset of feather buds elongation, the EF around chicken dorsal skin is relatively homogenous and inwardly

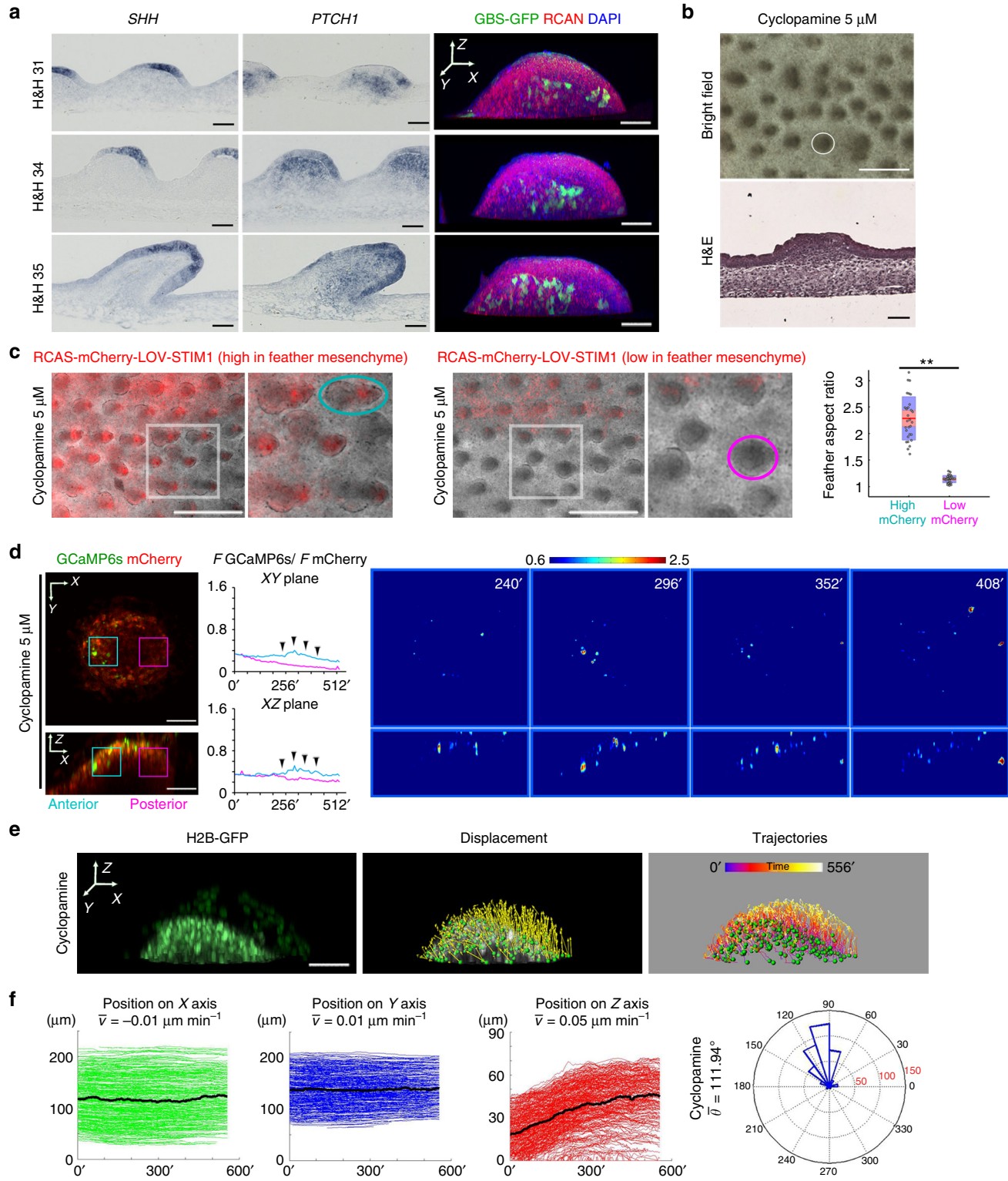

directed. Outward currents likely exist in other regions of the chicken embryo to complete the circuit on a global scale, which is known to occur in earlier chicken embryos[43]. These global circuits have also been detected in zebrafish embryos and Xenopus tadpoles[11,44]. Later on the current direction reverses locally at the anterior side of feather buds in chicken embryos. Another case of local electric current reversal is reported before limb bud formation in the Xenopus embryos[45]. Thus local circuit formation

may signify and participate in orchestration of local collective cell behaviors in different biological processes.

We postulate that the formation of these local electric circuits during feather bud elongation relies on active VGCC, CRAC, $Ca^{2+}$-activated $K^+$ channels, as well as the gap junction network. Previously, gap junction components have been detected in multiple types of skin appendages including feathers[46–48] and their networks are known to shape multicellular $Ca^{2+}$

**Fig. 7** SHH signaling underlies normal $Ca^{2+}$ oscillation and mesenchymal cell migration pattern in feather. **a** In situ hybridization reveals that *SHH* is expressed in the posterior–distal epithelium while *PTCH1* is enriched in the underlying mesenchyme. An RCAN virus-based GLI reporter demonstrates expansion of the mesenchymal SHH-responding zone anteriorly during feather elongation. Scale bar, 50 μm. **b** H&H 31 skins cultured for 4 days in 5 μM Cyclopamine exhibited dramatic inhibition of feather bud elongation (ellipse, $n = 8/8$). Scale bars, 500 μm, 50 μm. **c** Expression of opto-cCRAC in feather mesenchyme and cyclic blue light illumination partially rescued the Cyclopamine mediated inhibition of elongation by inducing new feather tips to form ($n = 3/3$). The buds with new tips exhibited more elongated morphology (cyan ellipse) than those poorly or not infected ones (magenta ellipse, $n = 29$ for each group, $**P < 0.01$, Wilcoxon rank test). Rectangles highlight magnified area. Scale bars, 500 μm. Customized boxplot: Mean (red) ± s.d. (pink), 95% confidence interval (violet). Dots denote individual data points. **d** Virtual sections of 4D $Ca^{2+}$ imaging of H&H 34 feather buds treated with 5 μM Cyclopamine ($n = 2$). Scattered $Ca^{2+}$ fluctuations were observed among the cells. Quantification of *F GCaMP6s/F mCherry* was conducted for the anterior (cyan rectangle) and posterior (magenta rectangle) mesenchyme and demonstrated in line plot. Scale bar, 50 μm. **e** 9 h 4D cell nuclear imaging of a feather bud treated with 5 μM Cyclopamine (mesenchyme highlighted in white). Scale bar, 50 μm. **f** Mesenchymal cell positions plotted over time and the averaged displacement angles ($n = 430$)

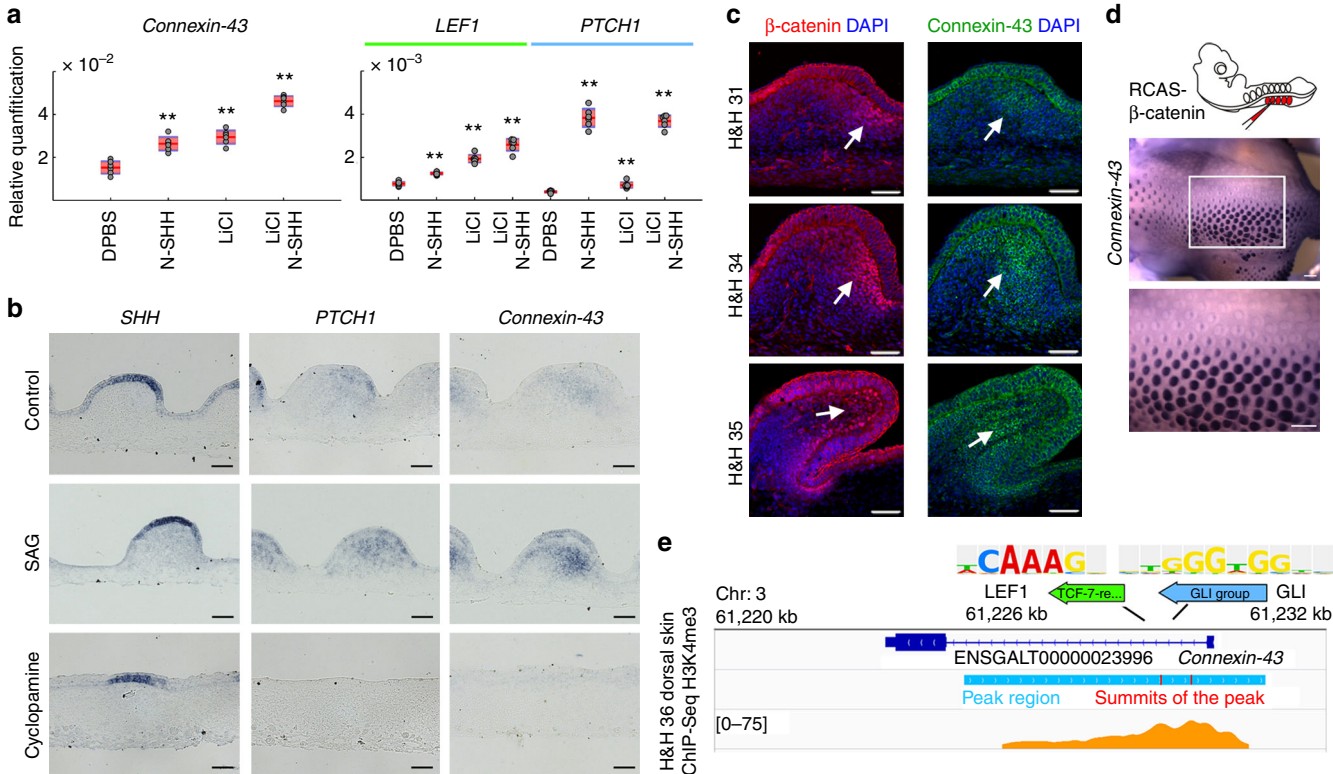

**Fig. 8** SHH and WNT signaling synergistically activate *Connexin-43* expression. **a** qPCR revealed a mild elevation of *Connexin-43* expression in cultured mesenchymal cells treated for 24 h with 0.25 μM SHH N-terminal peptide (N-SHH) or 7 mM LiCl to activate WNT signaling. The upregulation is more significant when both N-SHH and LiCl are present. Activation of SHH and WNT signaling are confirmed by *PTCH1* and *LEF1* qPCR, respectively ($n = 6$). $**P < 0.01$, two-sample Student's *t* test. Customized boxplot: Mean (red) ± s.d. (pink), 95% confidence interval (violet). Dots denote individual data points. **b** In situ hybridization demonstrated higher *Connexin-43* expression in feather buds treated with 500 nM SAG for 48 h. 5 μM Cyclopamine treatment decreased *Connexin-43* expression. Scale bar, 50 μm. **c** Immunostaining revealed close proximity of the Connexin-43 positive zone and nuclear β-Catenin zone in feather mesenchyme. Scale bar, 50 μm. **d** Mis-expressing constitutively active *β-Catenin* induced site-specific elevation of *Connexin-43* expression and abnormal feather polarity ($n = 5/5$). The region magnified is highlighted by rectangle. Scale bar, 500 μm. **e** Motif search in the active promoter region (Histone H3 Lysine 4 trimethylation, blue bar) of Connexin-43 identifies LEF1 and GLI-binding sites in close proximity. Orange peaks highlight enrichment levels

activities in a variety of excitable and non-excitable cells[49]. Due to their permeability to ions and small molecules, gap junctions serve as low-resistance pathways to spread ionic currents, and hence propagate membrane potential changes among neighboring cells[21,29]. The range of the gap junction network is regulated by diffusible morphogens from the biochemical signaling centers, including SHH diffused from distal placode epithelium[50] and WNT7A diffused from posterior bud epithelium[15]. Since both morphogens are capable of inducing *Connexin-43* expression, their overlapped area will be electrically coupled to the best degree. This is consistent with the fact that the $Ca^{2+}$ oscillations start at posterior–distal mesenchyme and gradually expand in area along the anterior–proximal direction. Meanwhile the expansion speed is slower than $Ca^{2+}$ wave propagation speeds reported in most other systems[51], implying $Ca^{2+}$ diffusion is not the rate-limiting step of this process. Additionally, in situ hybridization and in vitro qPCR data (Fig. 1c and Supplementary Fig. 12b) imply that SHH and WNT signaling likely also upregulate *STIM1* expression, which enables store-operated $Ca^{2+}$ influx through CRAC channels. In dissociated H&H 35 feather mesenchyme, we observed sparse cells with changes in cytosolic

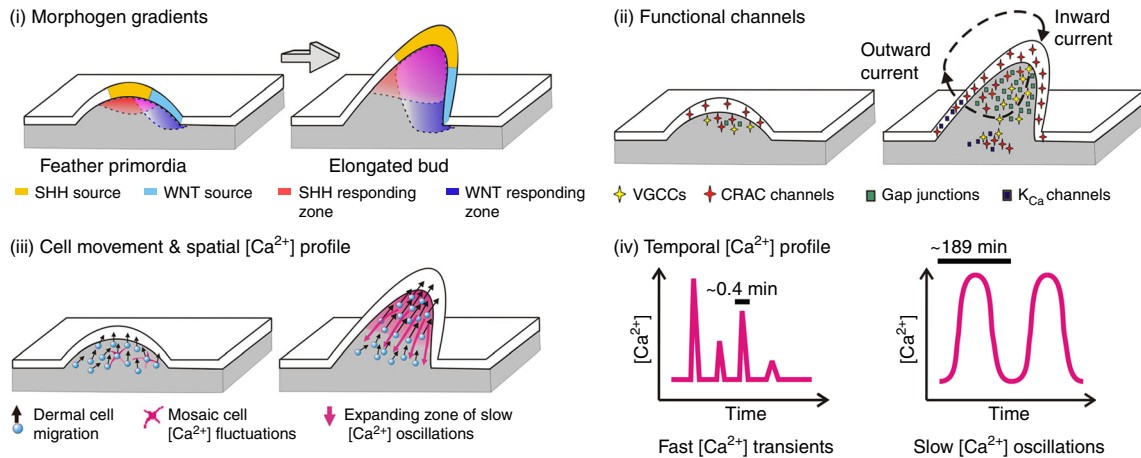

**Fig. 9** Biochemical and bioelectric signals guiding directed mesenchyme cell migration in feather. SHH and WNT7A protein produced in the epithelium diffuse into the underlying mesenchyme to induce the SHH and WNT-responding zone, respectively. Together they activate *Connexin-43* expression to compose intercellular gap junction channels. Heterogeneously distributed VGCCs and CRAC channels contributed to the inward current observed at posterior–distal part of elongating feathers, while activation of $Ca^{2+}$-activated $K^+$ channels may contribute to the outward current at anterior-basal part of feathers. The gap junction network establishes a low-resistance electrical path across feather mesenchyme, homogenizing differential $Ca^{2+}$ channel activities among cells to allow synchronized $Ca^{2+}$ oscillations to form. These slow oscillations coordinate the collective migration of mesenchymal cells. Meanwhile much faster $Ca^{2+}$ transients occur sporadically in individual cells

$Ca^{2+}$. Those cells that did show $Ca^{2+}$ changes typically exhibited sustained $Ca^{2+}$ elevations, implying continuous CRAC channel activation. Though sustained CRAC activity may depolarize membrane potential to activate the low-threshold voltage-gated T-type $Ca^{2+}$ channels, tissue-wide synchronized $Ca^{2+}$ oscillations were not observed until gap junction expression increased in mesenchyme. Thus we think gap junction mediated electric coupling is essential for synchronizing the originally heterogeneous VGCC and CRAC activation events, transforming sporadic $Ca^{2+}$ transients into organized oscillations. These oscillations facilitate the collective cell movements mediating feather elongation in the posterior–distal direction (Fig. 9).

$Ca^{2+}$ fluctuations have previously been implicated in modulating cell migration[52–54], convergent extension[51], apical constriction[55], etc. Our study demonstrates slow multicellular $Ca^{2+}$ oscillations coordinate collective mesenchymal cell migration in skin appendage organogenesis. We have three hypotheses of how these two events are linked: (1) Elevated cytoplasmic $Ca^{2+}$ promotes myosin-driven cell protrusions by activating myosin light chain kinase (promoting actomyosin contractility)[51]. (2) Elevated mitochondrial $Ca^{2+}$ (mitochondria are known to at least partially buffer cytoplasmic $Ca^{2+}$ increase) enhances ATP production[56]. (3) The oscillations serve as a communication mechanism enhancing sensitivity to gradients of chemotactic cues like FGF[57–60]. Since FGF signaling is required for normal *SHH* expression in feather[60], the chemotaxis and gap-junction-based communications may function in parallel.

Multicellular $Ca^{2+}$ oscillations are dependent on both gap junctions and CRAC channels in cells lacking VGCCs[61]. Our in situ hybridization data indicates that the *Connexin-43* expression region overlapped with that of *STIM1* in feather mesenchyme. Although CRAC channel inhibitor BTP2 did not block feather elongation (Supplementary Fig. 9), oscillatory activation of opto-cCRAC channels did enhance feather elongation (Fig. 4c). Beside VGCCs and CRAC channels, $Ca^{2+}$ transients could also be triggered by stretch-activated $Ca^{2+}$ channels[62], interestingly, the stretch-activated $Ca^{2+}$ transients observed in

human gingival fibroblasts are also contributed by VGCCs, suggesting a potential link between the two types of channels. Furthermore, mechanical stimulations could regulate the expression, subcellular localization and even phosphorylation state of Connexin-43[63]. There is a possibility that stretch between cells during feather mesenchymal migration helps reshape the functional gap junction network.

The mechanism described in this work is active when feather primordia ($< 100\,\mu m$ in length) elongate to become long feather buds ($> 200\,\mu m$ in length), representing a substantial change in the organ aspect ratio. Unlike the more stable adhesion molecule-mediated cell condensates, this mechanism enables transient cell collectives to form dynamically, effectively guiding mesenchymal cells during the construction of organ architectures. This study also demonstrates a proof-of-concept that manipulation of $Ca^{2+}$ patterns can modulate cell behaviors in tissues and organs located close to the body surface. Hence, the optogenetic $Ca^{2+}$ channels would have wide applications in procedures such as accelerating wound healing, decreasing scar formation, wound contraction, and inducing hair regeneration.

## Methods

**Egg resources**. Animal care and experiments were conducted according to the guidelines established by the USC Institutional Animal Care and Use Committee. White leghorn chicken eggs were purchased from Charles River (pathogen free) and AA laboratory.

**Vibrating probe measurement of feather bioelectric currents**. The vibrating probe technique for non-invasive measurement of endogenous electric current densities has been previously described[11]. The probe is an Elgiloy-stainless microelectrode (WPI #SSM33A70) coated with a thin layer of parylene insulation, leaving 1–2 μm of metal exposed at the tip. Using a nano-amp power supply, we electroplate a thin layer of gold and then a platinum ball onto the tip. The probe, mounted on a 3-dimensional micro-positioner (Line Tool Co., model H), is vibrated at high frequency (150–200 Hz) by a piezoelectric actuator in solution ~1 tip ball distance from the sample surface. If an electric current is present due to ion flux, the electric charge on the platinum ball fluctuates in proportion to the size of the current. The probe is connected to a lock-in amplifier (Stanford Research Systems, model SR530) that locks on to the probe's specific frequency. The probe is calibrated with a set current density of 1.5 μA/cm² at the start and end of experiment. Before measurements, the probe is vibrated in solution far from the sample (> 1 cm), where there is no electric current, to establish a baseline.

The probe is then moved into measuring position close and parallel to the sample surface, with the vibration direction perpendicular to the sample surface (see Fig. 1a). Measurements on different stages of chicken embryos were done at room temperature in chick Ringer's solution containing (mM): 148 NaCl, 5.6 KCl, 1.1 CaCl$_2$.6H$_2$0, 1 MgCl$_2$.6H$_2$0, 5 HEPES, pH adjusted to 7.0 with NaOH.

**Retrovirus preparation and injection**. RCAS-β-Catenin and RCAS-SHH have been reported previously[16,64]. To construct RCAS-GCaMP6s-T2A-mCherry, the GCaMP6s sequence was amplified from a CMV-GCaMP6s vector (Addgene # 40753), then digested with NotI, SpeI and subcloned into an RCAS(B)-T2A-mCherry vector. To construct RCAN-GBS-GFP vector, GBS-GFP was amplified from a pBS-KS vector from Dr. Briscoe (The Francis Crick Institute, UK), then digested with ClaI and subcloned into an RCAN vector. To construct RCAS-mCherry-LOV-cSTIM1, we first replaced the mouse STIM1 fragment (amino acid 336–486) in the original construct[32] with the corresponding chicken sequence amplified. Next the whole mCherry-LOV-cSTIM1 cassette was amplified with attB primers. The PCR fragment was subcloned into RCAS-DV (Addgene # 11478) using gateway recombination reactions. To construct the lentiviral vector expressing shRNA against Connexin-43, we designed the shRNA target sequence using Block-iT RNAi designer. The annealed primers were cloned into the pLL3.7 vector from Addgene (11795)[34]. Virus packaging was done in HEK293T cells with pCMV-VSVG and pCMV-dR8.2. Both RCAS and lentivirus were concentrated by ultra-centrifuging in a BECKMAN Coulter L8–80M (SW28 rotor) at 26000 rpm for 1.5 h. Concentrated viruses were injected into somites of H&H 14 or 17 chicken embryos. Primers used for subcloning are listed in Supplementary Table 2.

**Embryonic chicken skin explant/strip and cell culture**. For chicken skin explant cultures[14], dorsal skins from chicken embryos at corresponding stages were dissected in Hank's buffered saline solution and the explants were transferred to culture inserts (Falcon, 0.4 μm pore) for imaging. For making strips, the explants were transferred to PELCO glass-bottom dishes (50 × 7 mm) coated with 1 mg/ml Poly-L-lysine (Sigma, P6282) and Fibronectin (Sigma, F1141). After they fully attached the explants were examined under fluorescent microscope and the regions highly infected by RCAS-GCaMP6s-2A-mCherry virus were cut into one-bud-wide strips. The strips were placed in coated glass-bottom dishes on their sides for imaging. For dissociated mesenchymal cell culture, the epithelium and mesenchyme of embryonic chicken dorsal skins were separated by 20 min incubation in 2x CMF. The mesenchyme was dissociated by 0.35% type I collagenase treatment for 20 min. The digestion was stopped by the addition of FBS and the cells were seeded into culture dishes. Chemicals used in tissue culture: Nifedipine, Nitrendipine, Carbenoxolone disodium, 18-α-GA, Lanthanum chloride, Lithium chloride and Mefloquine were purchased from Sigma-Aldrich. N-[4-[3,5-Bis(trifluoromethyl)-1H-pyrazol-1-yl]phenyl]-4-methyl-1,2,3- thiadiazole-5-carboxamide (BTP2), [(3aR*,4 S*,7 R*,7aS)-1,3,3a,4,7,7a-Hexahydro-1,3-dioxo-4,7-methano-2H-iso-indol-2-yl]-N-8-quinolinylbenzamide (endo-IWR1), Phorbol 12-myristate 13-acetate (PMA), 3-Chloro-N-[trans-4-(methylamino)cyclohexyl]-N-[[3-(4-pyr-idinyl)phenyl]methyl]benzo[b]thiophene-2-carboxamide (SAG), and Thapsigargin were purchased from Tocris. Cyclopamine was purchased from LC Laboratories. Recombinant mouse Shh N-terminal peptide was purchased from R&D Systems. Feather bud aspect ratio (Major Axis/minor Axis) was calculated in ImageJ by using freehand or ellipse selection tool to mark the contour of feather buds.

**Characterization of cell proliferation and apoptosis**. Two hours before skin explants were collected and fixed in 4% PFA, 10 μl 1% BrdU (Sigma) were added to the culture media to label the proliferating cells. After fixation (4 °C overnight), samples were dehydrated in alcohol, cleared in Xylene, and then embedded in paraffin. The paraffin blocks were cut into 8 μm sections. Before staining for BrdU (Abcam ab6326 1:200 dilution), the sections were incubated in 10 mM Citrate buffer (pH = 6.0) at 95 °C for 0.5 h. Secondary antibody was anti-Rat-Alexa 488 (Thermo Scientific). To detect apoptotic cells we employed the TUNEL kit (Sigma 11684795910) and followed the protocol with proteinase K treatment (10 μg/ml for 15 min at room temperature).

**Time-lapse Ca$^{2+}$ imaging in cultured skin under perfusion**. H&H 34 or 35 embryonic chicken dorsal skin explants were cultured on a rectangular PET membrane (cut from Fisher Scientific 6-well cell culture insert) placed on two spacers in a WillCo 35 mm glass-bottomed dish (skin strips were directly put on coated glass-bottomed dish) for perfusion experiments. The culture media is DMEM (without Phenol red) containing 10% FBS, 2% chicken serum and 10 mM HEPES. Right before imaging, the culture media was replaced by Ringer's solution. The custom-built recording chamber was placed on a Zeiss Pascal confocal microscope with a 40 × /0.8 NA water immersion objective, and the chamber was warmed to 38.5 °C. Solenoid valve-controlled four barrel perfusion system was set up on the recording chamber, and the average flow rate of bath perfusion was about 1.4 ml/min. Due to the bath perfusion configuration, solution exchange was slow and was completed within 2 min.

**Time-lapse Ca$^{2+}$ imaging of dissociated cells**. Dissociated H&H 31 or H&H 35 embryonic skin mesenchymal cells were cultured on 0.01% poly-L-ornithine

coated-glass chips (0.5 × 0.5 cm). The mixture of 4 μM AM-Fura-2 Ca$^{2+}$ dyes (Life Technologies) and 0.04% pluronic F-127 (Sigma) were loaded in the presence of Ringer's solution for 25 min in a 37 °C CO$_2$ incubator. After washing with Ringer's solution for 10 min, each glass chip was placed onto a recording chamber equipped with a local perfusion system. The custom-built local perfusion system was placed right next to a field of view and was able to exchange each test solution with a time constant of 100 ms[65]. An inverted Axiovert 100 microscope (Zeiss) was used and equipped with a monochromator (Polychrome V; TILL Photonics) to generate 350 nm and 380 nm excitation wavelength for Fura-2. Each excitation wavelength was illuminated for 200 ms at 1 Hz sampling frequency through a 63 × /0.9 NA achroplan water immersion objective. The corresponding emission intensity was detected with a 16-bit depth EMCCD camera (iXon Ultra 897, Andor Technology) operated by MetaFluor (Version 7.8, Universal Imaging). All recordings were done at room temperature. For ROI analysis, background-subtracted intensity traces were imported into IGOR Pro 6.22 A (Wavemetrics), and analyzed using custom-written data analysis routines. For testing activities of the chicken version of the opto-cCRAC construct, Hela cells were transfected with 300 ng plenti-mcherry-LOV-cSTIM1 and 300 ng pGP-CMV-NES-jRCaMP1b (addgene#63136) in 35 mm glass-bottom dishes. Sixteen hours post-transfection, cells were recorded under 594 nm channel: 30s-off, followed by two cycles of on (blue light 1 min) and off (2 min).

**Solutions used for time-lapse Ca$^{2+}$ imaging**. The skin explants expressing RCAS-GCaMP6s-T2A-mCherry were washed with 2 mM Ca$^{2+}$ Ringer's solution containing (mM): 140 NaCl, 2.8 KCl, 10 HEPES, 1 MgCl$_2$, 2 CaCl$_2$, and 10 D-glucose (pH 7.2 adjusted with 1 N NaOH, 290~ 300 mOsm adjusted with D-glucose). 100 mM KCl solution was composed of 100 KCl, 50 NaCl, 10 HEPES, 1 MgCl$_2$, 2 CaCl$_2$, and 10 D-glucose (pH 7.2 adjusted with 1 N KOH, 290~ 300 mOsm adjusted with D-glucose). Ca$^{2+}$ free solution was composed of 140 NaCl, 2.8 KCl, 10 HEPES, 1 MgCl$_2$, 1 mM EGTA, and 10 D-glucose (pH 7.2 adjusted with 1 N NaOH, 290~ 300 mOsm adjusted with D-glucose). All the solutions were filtered using bottle-top filters with a 0.2 μm pore size membrane before use.

**Patch clamp experiment for VGCC current recording**. Dissociated mesenchymal cells from H&H 35 embryonic chicken skins were cultured on collagen-coated-glass chips (rat tail type I collagen, Sigma). Internal solution was composed of (in mM) 120 CsGlutamate, 10 HEPES, 2 MgATP, 0.3 Na$_2$GTP, 10 NaCl, 10 EGTA, and 10 TEACl (pH 7.2 adjusted with 1 N CsOH, 290 mOsm). The internal solution was filled into fire-polished glass pipettes prepared using a PC-10 patch pipette puller (Narishige, Japan). Pipette resistance ranged from 4.5 Mohm to 5.5 Mohm. The cells were placed on a recording chamber and washed with 2 mM Ca$^{2+}$ Ringer's solution. An inverted Axiovert 100 microscope (Zeiss) with a 63 × /0.9 NA water immersion objective was used to locate single mesenchymal cells. After forming a gigaohm seal between a patch pipette and cell plasma membrane, a standard whole-cell configuration was prepared using an EPC9 amplifier (HEKA Elektronik, Germany). Pipette capacitance and membrane capacitance were compensated and further eliminated using a P/4 protocol during current recordings. Series resistance ranged from 10 Mohm to 20 Mohm. A step voltage protocol written using PatchMaster Version 2 × 73.5 was applied to voltage-clamped cells. Currents were filtered at 2.5 kHz using a 4-pole bessel filter and digitized at a sampling frequency of 10 kHz. For data analysis, current traces were imported into IGOR Pro 6.22 A (Wavemetrics), and current-voltage plots were generated using a custom-written data analysis routine.

**4D Ca$^{2+}$ imaging and cell nucleus imaging**. H&H 34 chicken dorsal skins were cultured for 3–4 h before imaging in a Fisher Scientific 6- well culture insert (upper wall trimmed) mounted on 4 spacers glued to a WillCo 50 mm glass-bottom dish with 3 ml culture media underneath it. Right before imaging 1.8 ml culture media was added on top of the skin. A non-lubricated condom was used to make a sealed chamber between the dish and the microscope lens. Both the lens and the dish were warmed for 1.5–2 h before imaging. For 4D ratiometric Ca$^{2+}$ imaging in skin explants, the interval between z-stack scans was 6 min (scanning time was ~ 2 min). The interval was 1.5 min for skin strips. For 4D nuclear imaging, the interval was 3 min (~ 1 min scanning time). 3D rendering, cell tracking was done in IMARIS (Bitplane). Segmentation of the feather mesenchyme, ROI analyses were done in ImageJ. Quantification of movement patterns and generation of pseudocolor images were done in MATLAB. The duration of the slow Ca$^{2+}$ transients were determined by measuring the time interval between the peaks or valleys on the F GCaMP6s/F mCherry ratio plots. To calculate the spreading speed of the multicellular Ca$^{2+}$ transients, the frontiers of the wave propagation were manually determined in the psendo-color images at different time points and the distances between the frontiers were measured and divided by the time intervals.

**Photo-activation of opto-cCRAC in cultured cells and skin**. For the cultured mesenchymal cells, concentrated RCAS-mCherry-LOV-cSTIM1 virus was added to cell culture media (1:200) for 48 h. The cells were loaded with the Ca$^{2+}$ indicator AM-Fluo-4 plus pluornic F-127 (Thermo Fisher) for 30 min and then washed with HBSS (no Phenol Red) before imaging. Imaging was done in a Zeiss LSM510 confocal microscope at 38 °C. For the skin culture, skins from H&H 31 embryos

injected with the virus (only to the left body side) were cultured on a culture insert for 48 h within a black WillCo dish with a hole on top to insert LED light. Photostimulation was provided by an external blue light (470 nm, at roughly 20 mW/mm$^2$, ThorLabs Inc., Newton, NJ, USA). Light cycles (2 min on/off) were applied by programming DC2100 LED Driver with LabVIEW (National Institute). Measurement of feather aspect ratios were calculated in ImageJ by using the elliptical selection tool to mark feather contour.

**RNA-Seq and ChIP-Seq.** Skins dissected from H&H 31 and H&H 35 chicken embryos were used for RNA-Seq and ChIP-Seq library preparation. For RNA-Seq epithelial and mesenchymal tissues were separated in 2X CMF (calcium-magnesium free saline). Total RNA was extracted using Trizol reagent. For library construction we used TruSeq RNA Sample Prep Kit Version2 (Illumina). Two biological replicates were sent for sequencing (50 bp, single end) with Illumina HiSeq 2000 in the USC Epigenome Center. FastQ files were trimmed and mapped to the chicken genome (galGal4) using Partek Flow. Further analysis was done in Partek Genomic Suite. For the ChIP-seq experiments[66], embryonic skin were dissociated by 0.35% collagenase (type I) digestion before formaldehyde cross-linking. Sonication time was 20 min (30 sec on/off cycle). H3K4me3 antibody came from Abcam (ab8580, lot GR188955-1). IgG and input DNA were used as background controls. Biological replicates were sent for library preparation and sequencing by the USC Epigenome Center. Reads were trimmed based on quality and aligned to galGal4 in Galaxy. Peak calling was done with QuEST[39]. For motif search we applied TRANSFAC (BIOBASE), with the cut-off condition set to minimize false positives.

**In situ hybridization and immunostaining.** In situ hybridization and immunostaining were done as previously described[14]. Primers used for cloning in situ probes are listed in Supplementary Table 2.

Primary antibodies for immunostaining: β-Catenin (Sigma, 15b8); Connexin-43 (Santa Cruz Biotechnology, sc-9059); BrdU (ABCAM, AB6326); GFP (NOVUS, NB600-308); LCAM (Chuong and Edelman. 1985); Neurofilament (3A10, DSHB); VIM (H5, DSHB). Secondary antibodies were labeled with Alexa Fluor 488 or 546 (Thermo Fisher).

**RT-qPCR for skin and cultured mesenchymal cells.** RNA extraction was done with Zymo Research Direct-zol RNA Kits. Reverse transcription was done using Superscript III First Strand Synthesis kit. The RNA and cDNA concentrations were measured with the NanoDrop 2000 spectrophotometer and normalized between samples. Primers used for qPCR are listed in Supplementary Table 2. The Ct values were measured using the Agilent Mx3000P qPCR system. The relative quantification was done by pyQPCR Version 0.9 software.

**Scrape-Load dye transfer assay.** Dorsal skins of H&H stage 30 and 33 chicken embryos were peeled off and cultured on 6-well culture inserts for 1 day till they reached the proper developmental stages. Control, 18-α-GA, or PMA-treated skins were covered with 2 mg/ml Lucifer Yellow (Sigma) and 1 mg/ml Rhodamine–Dextran (Thermo Fisher) dissolved in HBSS when the feather buds were cut by Tungsten needles. Dyes were rinsed off after 5 min and the skins were washed two more times. Images of the scraped feather buds were taken 8–10 min after dye loading.

**Statistics.** In order to test whether datasets were normally distributed random samples, the Kolmogorov–Smirnov tests were conducted. Two independent sample T-tests were used for comparing unpaired sample groups. For some datasets not equally and normally distributed, Wilcoxon rank tests were conducted using IGOR Pro or MATLAB to evaluate statistically significant difference between two samples. Chi-square test was conducted in MATLAB. To evaluate whether the directionality of migrating mesenchymal cells (circular dataset) had significant differences in different experimental conditions, Watson's $U^2$ tests were conducted in MATLAB.

## Data availability

RNA-Seq data have been deposited in the NCBI GEO database under accession code GSE86251. The ChIP-Seq data have been deposited in the GEO database under accession code is GSE122049. The authors declare that all other data supporting the findings of this study are available within the article and its Supplementary Information files, or are available from the authors upon request.

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

## Acknowledgements

C-.M.C., A.L., R.B.W. and P.W. are supported by National Institute of Arthritis and Musculoskeletal and Skin Diseases (NIAMS) R01-AR47364, AR60306, GM125322. A.L. is also supported by California Institute of Regenerative Medicine (CIRM) training grant TG2-01161 and Doerr Stem Cell Challenge Grant. J-.H.C., R.H.C. are supported by NIH BRP 5R01EY022931. B.R., M.Z. are supported by grants from AFOSR (FA9550-16-1-0052), NIH (1R01EY019101), NEI Core Grant (P30 EY012576, to J.S. Werner) and Research to Prevent Blindness. L.H., T.P., Y.Z. are supported by NIH R01-GM112003, R21-GM126532 and the Welch Foundation. We thank Dr. James Briscoe (Francis Crick Institute, United Kingdom) for providing the GBS-GFP plasmid. We thank Drs. Tingxin Jiang, Ya-Chen Liang, Jie Yan, Mingxing Lei, Masafumi Inaba, and other Chuong Lab members for their support. We thank USC Stem Cell Microscopy Core Facility and the Cell and Tissue Imaging Core, USC Research Center for Liver Disease for assistance in imaging (NIH P30DK048522); USC Epigenome Center for sequencing; Yibu Chen and Meng Li of the USC Norris Medical Library Bioinformatics Service for assistance in RNA-Seq and ChIP-Seq analysis.

## Author contributions

A.L., J-.H.C., C-.M.C., and R.H.C. conceived the overall experimental designs. M.Z., B.R., and C-.M.C designed the vibrating probe assay and B.R. conducted the experiments. J-.H.C. and A.L. conducted and analyzed time-lapse $Ca^{2+}$ imaging data obtained from skin explants and strips. J-.H.C. and R.H.C. conducted single cell $Ca^{2+}$ imaging, immunostaining, patch clamp experiment and analyzed the data. C-.Y.Y. wrote an intensity modulated display analysis routine to generate ratiometric $Ca^{2+}$ images. A.L. conducted the 4D $Ca^{2+}$ imaging, 4D cell nuclear imaging. Y.L., R.B.W., and A.L. analyzed the 4D data. C-.C.T and A.L. conducted the scrape-loading dye transfer assay. L.H., P.T., Y.Z., and A.L. designed the opto-cCRAC experiments and A.L. conducted the experiments. P.W. did the RNA-Seq, ChIP-Seq sample preparation and A.L. did analysis of the data. A.L. designed the shRNA for pLL3.7 and P.W. did the cloning. A.L. conducted the in situ hybridization, immunostaining experiments. A.L., J-.H.C., R.B.W., R.H.C., and C-.M.C. contributed to manuscript writing and editing. This is a multi-disciplinary study. C-.M.C. contributed more on feather bud morphogenesis and R.H.C. contributed more on ion channel and $Ca^{2+}$ signaling.

## Additional information

**Competing interests:** The authors declare no competing interests.

