## [Peer Review File · Nature Communications]

Reviewers' Comments:

Reviewer #1:

Remarks to the Author:

This paper documents roles for calcium signalling of collective migration underlying the morphogenesis feather buds in chick embryo dorsal skin explants. A key component of the transformation of feather buds is their elongation and bending, a process that is critically dependent on differential migration of mesenchymal cells underlying the ectoderm. As in many systems a key question is what mechanisms might control the coordination of these collective migrations. The authors decided to investigate the hypothesis that propagating electrical signals could be component of the guidance mechanism.

First of all they use a vibrating probe to measure changes in extracellular electrical fields and they report a pronounced changes during feather bud development. They perform an expression analysis to look for expression of voltage gated calcium channels as well as regulators of cytosolic calcium (CRAC) channel. The rationale being that calcium transients have often been associated with cell movement. They also find evidence for expression of various gap junction components both in the ectoderm and mesenchyme that may be involved in electrical coupling of cells. To measure calcium transients they use both genetically encoded calcium detectors for long term calcium imaging in explants as well as the more conventional calcium sensitive ratio-metric dyes. To manipulate calcium channels and gap junction they make use of various relatively well understood and specific small molecule chemical inhibitors as well as a photoactivatable endoplasmatic reticulum calcium sensor STIM1. Furthermore they perform a series of patch-clamp experiment in isolated cells to validate the expression of a variety of channels. Using this impressive array of techniques they undertake a well documented and clearly described study of involvement of calcium signalling in feather bud morphogenesis.

The main conclusions are that calcium transients can be detected in the mesenchymal cells of developing feather buds. These transients require voltage gated calcium channels and are less dependent on the CRAC channels, but they do require electrical coupling of mesenchymal cells through functional gapjunctions. Inhibition and activation experiments shown that inhibition of these calcium transients results in abnormal movement of mesenchymal cells resulting in defective feather bud elongation and morphogenesis. It is furthermore shown that Hedgehog and Wnt signalling that are known to play feather bud morphogenesis affect the expression of a number of key components involved in the electrical coupling of the cells and observed calcium transients. These experimental results are all well documented and the data appear convincing and are well described. They use a state of the art methodology and some exciting new tools such as the light-activated calcium sensor STIM1. They show undoubtedly that electrical coupling and or calcium transients play a key role in the collective cell migration of the mesenchymal cells during feather bud morphogenesis. These data therefore certainly highlight a potentially new and exciting components in the control of collective cell migration and certainly deserve publication.

The point that does not become very clear is what role the calcium transients/ electrical coupling play in the differential control of feather bud mesenchymal cells. Key unanswered questions are whether these electrical signals are instructive in directing migration or a necessary component of the machinery. If they are instructive what are the mechanism through which they act?

I have a specific questions and remarks below

1: The vibrating probe experiments seem to show all inward directed currents in the early buds. Electrical circuits need to be closed meaning that the integral of the inward and outward currents must match. Why are no outward current detected?

2: Are any current fluctuations measured during the calcium transients, i.e. is there any evidence that these calcium oscillations are coupled to large scale membrane depolarisations? It could be instructive to have some measurements of membrane potential using voltage sensitive dyes, although this would be a considerable amount of work.

3: Although the authors speak of calcium oscillations there are really only a few transients. I guess

it is possible that there are faster oscillations but they may be difficult to detect since this would require fast imaging.

4: in the experiments with the photoactivatable STIM1 is there a response when the cells are not illuminated, i.e. is it clear that the construct has no detectable activity in the absence of blue light illumination?

5: An important question is whether the calcium signals/depolarisation propagates in a directional manner which would be required for them to give any spatial information. There is a slight hint in the movies that this might be case but a more careful analysis of the spatio-temporal aspects of these transients might show whether this is the case. It would require averaging results from a variety of experiments.

6: If possible it would be desirable to correlate the dynamics of the calcium transients with changes in the dynamics and direction of cell migration. This could be done by using the mosaic expression of the calcium indicator as marker for cell tracking.

7: it might be interesting/useful for the authors to speculate on how they think that the coupling calcium transients could direct cell migration and how this would account for in the observed bud morphogenesis

Reviewer #2:

Remarks to the Author:

The manuscript entitled "Coupling of Biochemical-Bioelectric Signals Mediates Collective Dermal Cell Migration" by Li et al. studies the role of Ca²⁺ dynamics in the collective migration of mesenchymal cells during feather development in the chick. By live imaging and pharmacological and genetic manipulations of signaling molecules the authors find that voltage-gated Ca²⁺ channels and gap junctions are necessary for oriented collective cell migration and appropriate feather formation. Morphogenetic proteins Shh and Wnt influence this signaling by regulating expression of gap junction molecule Connexin 43. The authors conclude that biochemical and "electrical" signals are intermingled to imprint a spatiotemporal map to the morphogenesis of tissues during development.

The study of the features of Ca²⁺ signaling that are important for tissue morphogenesis is interesting and relevant for many fields including developmental biology, cell biology and cancer. However, I found major issues with the experimental design and the conclusions that the authors extract from the presented results.

Major concerns

1. Some of the RNAseq data does not seem to match up with the in situ hybridization data shown in Fig. 1. For instance, Connexin 43, which appears to be relevant for the findings presented in the manuscript, shows an upregulation in mesenchymal cells in RNAseq data when transitioning from stage 31 to stage 35 but the in situ hybridization images do not reflect that change. Also, CACNA1C is only shown in RNAseq for mesenchymal cells and not in epithelial cells, although in situ show robust expression in epithelia and developmental upregulation in epithelia, which doesn't match with the lack of KCl-induced Ca²⁺ response in epithelial cells. These discrepancies are not discussed by the authors.

2. Are the feather tissues innervated? Are there sensory terminals? (Hemming et al., 1994). The expression of channels may come from neural structures. In situ hybridizations (Fig. 1 and Suppl Fig. 1) do not provide the spatial resolution to address that, nor they are a good reflection of the expression of functional channels. Immunohistochemistry and assessment of colocalization with cell identity markers could provide more meaningful information on the specific expression of these channels in different cells.

3. The response to high KCl seems too slow to come from influx through opening of voltage-gated

Ca²⁺ channels. Perhaps neural projections are releasing some signal that in turn induces Ca²⁺ transients in mesenchymal cells?

4. It is unclear how the cell cultures compare to explants in terms of cell type composition. It is possible that different Ca²⁺ responses obtained upon addition of KCl are due to a mix of mesenchymal and epithelial cells in culture. The authors should have some reporter of cell identity or stain cells post imaging to understand what type of cell gives which response.

5. The paradigm that the authors used for membrane depolarization consists in adding 100 mM KCl extracellularly. This is not a physiological manipulation. What would be the physiological signal that would depolarize mesenchymal cells?

6. It seems that the level of expression of mCherry-GCaMP6 is always lower in epithelial compared to mesenchymal cells, based on the intensity of the red and green signal in both cell types, which also results in lower ratio of GCaMP6 fluorescence/mCherry fluorescence. Hence it is unclear whether the different responses and Ca²⁺ activity observed in both cells is due to a technical issue of differences in transfection/expression or distinct Ca²⁺ signaling.

7. PMA is not specific enough to conclude on involvement of gap junctions when the drug is utilized in several experiments presented in this manuscript. The results with PMA should be either removed from the paper or left in supplementary material. Moreover, differences in phenotype elicited by PMA and carbenoxolone argue that PMA is hitting on other targets, and it is used at an excessive concentration of 500 nM. It seems that the authors favored the PMA results over the carbenoxolone ones because the former were more dramatic/significant but this is misleading because we are looking at a drug that has a plethora of targets, hence the results become very inconclusive in terms of the role of specific Ca²⁺ signaling molecules.

8. Unfortunately, carbenoxolone is also a "dirty" drug (Connors BW, Epilepsy Curr, 2012) and it doesn't help that the authors use 150 μM, which seems excessive. Experiments with genetic inhibition of gap junction/connexin43 function are desirable. There are also more specific inhibitors for gap junctions and connexin 43 that the authors could use.

9. Figure 2g shows that nifedipine, an L-type voltage-gated Ca²⁺ channel blocker blocks completely the change in [Ca²⁺]_i when KCl is added, suggesting that these channels are responsible for the full response to high KCl. This is not what the authors conclude on. The authors need to revise/clarify this mismatch.

10. I am confused about the decrease in [Ca²⁺]_i that the authors report in vitro upon KCl addition in the majority of cells. The authors argue that there is a constitutive CRAC-dependent influx that is sensitive to depolarization. However, CRAC channels are voltage-independent, and although they are believed to be inhibited by intracellular [Ca²⁺] increases, the fact that these traces show no increase in [Ca²⁺]_i argue for an artifact that results in this drop in GCaMP6 signal rather than any real biological explanation. For instance, a change in focal plane can result in a drop in fluorescence intensity. Moreover, why isn't this response present in explants?

11. Electrophysiological recordings were done only in cultured cells. It is unclear what is the cellular composition of these cultures. What is the percent of mesenchymal cells? And epithelial? Primary cultures are never 100% pure. Hence, it is unclear how these results correlate with the ex vivo or in vivo model used in this study.

12. The measurements of direct currents through VGCCs are reported to be from n=3/8. What does this mean? 3 out of 8 cells showed these currents? 3 out of 8 showed the reported I-V plot? How does this incidence, determined in vitro, correlate with in vivo studies? What is different about the 3 "successful" cells compared to the other 5?

13. Moreover, the authors state that the I-V plot corresponds to T-type Ca²⁺ channels, however, only L-type VGCC blockers are used throughout the paper. This is confusing and suggest that the in vitro and explant systems that the authors used do not completely agree with each other, questioning the validity/relevance of the in vitro data.

14. "Contrary to our expectation that CRAC channels are crucial for mesenchymal cell migration, the most potent CRAC inhibitors failed to block feather elongation (Supplementary Fig. 7). Yet artificial Ca²⁺ oscillations triggered by photoactivatable opto-CRAC channels enhanced feather elongation". This sentence is confusing but the results themselves are not. Doesn't the lack of phenotype from treating samples with CRAC inhibitor rule out CRAC channels' role/importance in feather elongation?

15. In Figure 4c, when half of the embryo was transfected with STIM1, the STIM1-expressing feathers look more like the controls than the "wild-type" feathers in the same animal. Is there a compensatory mechanism? Is it non-cell/feather autonomous? Also, while Fig. 4d,f shows quantitative analysis of feather elongation when STIM1 or GCaMP6 are transfected, this analysis is not provided for drug-treated samples or controls in 4b. Hence, it is difficult to compare different experimental groups and the significance of the results. The same is the case for Suppl Fig. 7 where quantitative analysis is not provided.

16. Additionally, is the more rounded feather shape seen in the control of Fig. 4c the same phenotype as for carbenoxolone-treated samples shown in Suppl Fig. 7? Perhaps the phenotype is more a developmental delay rather than a real impairment of feather polarity/morphogenesis?

17. In Fig. 7d a control image of Connexin 43 expression pattern in a wild-type sample is missing. Otherwise, it is not possible to conclude that β -catenin overexpression leads to higher Connexin 43 expression as the authors do.

18. There are several instances of repeated results between figures and suppl figures (i.e., Fig. 4 and Suppl Fig. 7; Fig. 5 and Suppl Fig. 8). The authors should avoid redundancy.

19. The authors need to rule out that what they report as changes in cell migration are not a consequence of changes in cell density due to altered proliferation or cell death by performing proliferation and apoptosis assays in control and experimental groups.

20. The authors should define/explain why they call the transients in Suppl Fig. 9 Ca²⁺ spikes unlike in the rest of the manuscript where they call them transients, oscillations. It is quite confusing. Are there different types of transients?

21. Have the authors look at more acute responses to SAG or Shh than 20 min after? Are there changes in the frequency of Ca²⁺ transients with Shh/SAG treatments?

22. In many experiments the n of samples is equal 2. This is not acceptable and it is mostly in experiments that seem more physiologically relevant in their design. The authors need to increase the n of these experiments to make a valid conclusion (Figs. 4, 5, 6; Suppl. Figs. 5, 8; Suppl. videos 21-28, 31).

23. In the model/summary Figure 8 the authors state: "Heterogeneously distributed VGCCs and CRAC channels contributed to the inward current observed at posterior-distal part of elongating feathers, Ca²⁺ activated K⁺ channels may be elevated by high tissue Ca²⁺ levels and contributed to the outward current at anterior-basal part of feathers". There is no evidence provided by the authors on either the heterogeneous distribution of VGCCs and CRAC nor the Ca²⁺-activated K⁺ channels or the outward current at anterior-basal part of feathers. Hence, this is over speculative and the authors should stay closer to what their results provide in their model.

24. The authors claim in the opening paragraph of the Discussion that “To our knowledge, the discoveries here are the first report of slow, multicellular synchronized Ca²⁺ oscillations to coordinate the collective cell movement patterns in tissue morphogenesis”, but in fact many reports have demonstrated this in different contexts. To name a few, it has been shown that multicellular Ca²⁺ transients are present during neurulation and are necessary for neural tube morphogenesis (Suzuki et al., Development 2017) and that bidirectional radial Ca²⁺ activity regulates neurogenesis and migration during early cortical column formation (Rash et al. Science Advances 2016). The authors should cite previous studies (others are: Ellison et al, PNAS 2015; review: Markova and Lenne, Seminars in Cell and Developmental Biology 2012) and revise their discussion. The originality of this study is somewhat overstated.

Minor concerns

1. In page 15, 3rd sentence from top should refer to Supplementary Fig. 10b, not “11b”.
2. In page 15, 7th sentence from bottom should refer to Supplementary Fig. 10c, not “10b”.

Reviewer #3:

Remarks to the Author:

In this manuscript Li et al. test the hypothesis that calcium signaling dynamics mediate coordinated mesenchymal cell migration using the developing chick feather bud as their main experimental model. It is well known that calcium signaling is a critical mediator of multiple developmental processes however specific mechanisms for how calcium signals are propagated and their functional significance have been elusive, primarily due to the extreme dynamic nature of calcium and ion currents in live cells and tissues. In this study the authors use a robust and previously tested explant model system and a range of powerful molecular and optogenetic approaches coupled with multi-dimensional live imaging to study the role of calcium oscillations during feather development. Overall, the novelty, breadth and stringency of the experimental approaches and the broad biological implications of the findings make this manuscript appropriate and valuable for the general readership of Nature Communications.

As a general comment I find the feather explant model the authors have previously developed and validated to be exceptionally useful for its biological relevance, genetic amenability and suitability for live imaging approaches. The quality of the live imaging data throughout the study is very good and the various reporters used to visualize calcium and other cell activities have been previously validated and appropriate.

However, the authors refer to the epithelium and mesenchyme of the feather bud throughout the manuscript without clearly defining the two populations molecularly or even morphologically. This is problematic when evaluating the live imaging data, in the absence of secondary markers. Although it is intuitive to distinguish the two populations by location within the tissue structure or by referring to the literature it would be useful that the authors more clearly define them, perhaps in a Fig1 subpanel to assist the non-expert reader and remove any ambiguity. This is especially important for the mesenchyme that by nature is thought to be a heterogeneous population in multiple tissues.

The data in Figure 4 are convincing and clearly define the significance of this study in defining a specific role of calcium signaling in feather bud development. Subsequent figures provide some possible insight on the underlying molecular mechanism that regulates calcium currents and role of Hh signaling. However, in these experiments (figure 6) it is unclear if asynchronous calcium oscillations are the primary effect of the impaired SHH or the secondary effect of other cell interactions in the explant. Subsequent studies are needed with cell type specific knockdown of SHH signaling and rescue to show that this is affecting movement of mesenchymal cells directly but these are clearly beyond the scope of this manuscript.

In the culture experiments of figure 3 it is not really clear how these cells were cultured and while I believe the claims about the data, these results could easily be artefactual. They could be a result of dysregulation resulting from dissociation or a number of other reasons. Moreover, the mesenchymal population itself is very likely heterogeneous, not simply their response to KCl. If I am reading this correctly, the contribution of this to the paper as a whole is that mesenchymal cells at this stage are a heterogeneous population, implying that because they respond differently to depolarization their connectivity is essential to their functioning as a group to mobilize in a directed pattern. The authors may want to use the RNA-seq data to determine whether there are subpopulations of mesenchymal cells here and mention that they are marked by different expression of these calcium related genes. However, this may also be beyond the scope of this paper but a more thorough interpretation of the data and the caveats relating to fig 3 may need to be discussed in the manuscript.

Minor comment: Reference 16 in the text appears to be citing the wrong paper.

Overall the data presented in this manuscript significantly advance the field and our understanding of the role of ion currents in regulating developmental growth and patterning processes and may serve as an important stepping stone for further in vivo studies to fully resolve the underlying genetic and molecular mechanisms. As such I recommend that this manuscript is accepted with minor changes based on my comments above.

Reviewer #1 (Remarks to the Author):

This paper documents roles for calcium signaling of collective migration underlying the morphogenesis feather buds in chick embryo dorsal skin explants. A key component of the transformation of feather buds is their elongation and bending, a process that is critically dependent on differential migration of mesenchymal cells underlying the ectoderm. As in many systems a key question is what mechanisms might control the coordination of these collective migrations. The authors decided to investigate the hypothesis that propagating electrical signals could be component of the guidance mechanism.

First of all they use a vibrating probe to measure changes in extracellular electrical fields and they report a pronounced changes during feather bud development. They perform an expression analysis to look for expression of voltage gated calcium channels as well as regulators of cytosolic calcium (CRAC) channel.

The rationale being that calcium transients have often been associated with cell movement. They also find evidence for expression of various gap junction components both in the ectoderm and mesenchyme that may be involved in electrical coupling of cells. To measure calcium transients they use both genetically encoded calcium detectors for long term calcium imaging in explants as well as the more conventional calcium sensitive ratio-metric dyes. To manipulate calcium channels and gap junction they make use of various relatively well understood and specific small molecule chemical inhibitors as well as a photoactivatable endoplasmic reticulum calcium sensor STIM1.

Furthermore they perform a series of patch-clamp experiment in isolated cells to validate the expression of a variety of channels. Using this impressive array of techniques they undertake a well documented and clearly described study of involvement of calcium signaling in feather bud morphogenesis.

The main conclusions are that calcium transients can be detected in the mesenchymal cells of developing feather buds. These transients require voltage gated calcium channels and are less dependent on the CRAC channels, but they do require electrical coupling of mesenchymal cells through functional gap junctions. Inhibition and activation experiments shown that inhibition of these calcium transients results in abnormal movement of mesenchymal cells resulting in defective feather bud elongation and morphogenesis. It is furthermore shown that Hedgehog and Wnt signalling that are known to play feather bud morphogenesis affect the expression of a number of key components involved in the electrical coupling of the cells and observed calcium transients.

These experimental results are all well documented and the data appear convincing and are well described. They use a state of the art methodology and some exciting new tools such as the light-activated calcium sensor STIM1. They show undoubtedly that electrical coupling and or calcium transients play a key role in the collective cell migration of the mesenchymal cells during feather bud morphogenesis. These data therefore certainly highlight a potentially new and exciting components in the control of collective cell migration and certainly deserve publication.

Reviewer #1-Question 1: The point that does not become very clear is what role the calcium transients/ electrical coupling play in the differential control of feather bud mesenchymal cells. Key unanswered questions are whether these electrical signals are instructive in directing migration or a necessary component of the machinery. If they are instructive what are the mechanism through which they act?

Reply:

We feel that the calcium transients/electrical coupling can play both an instructional role and also function as necessary components which mediate collective cell migration within feather buds.

We appreciate that the reviewer highlighted this critical point and indicated that our data “show undoubtedly that electrical coupling and or calcium transients play a key role in the collective cell migration of the mesenchymal cells during feather bud morphogenesis. These data therefore certainly highlight a potentially new and exciting components in the control of collective cell migration...”. This greatly encourages us to continue to elucidate “...detailed control and mechanisms of electrical coupling and the calcium transients in the differential control of feather bud mesenchymal cells” as a new focus of our next research. One way to answer the differential roles of calcium signals, electrical coupling and large-scale standing currents along a single feather bud may be to measure endogenous currents using the vibrating probe in the conditions that calcium channels, and/ or gap junctions are manipulated genetically and pharmacologically. We agree with the reviewer about a potentially critical and novel “electrical coupling” mechanism in collective behavior control and our data in this manuscript set a solid basis for the next phase of our research. Furthermore, we are also very intrigued by the fact that the electric field applied to a cell in vitro can affect directional migration in isolated cells through interaction between K^+ channels and polyamines (Nakajima, K. et al. Nature communications 6, 2015). We speculate that standing currents generated by polarized channel expressions in a bud may redistribute or localize charged molecules such as PIP2/3 following the endogenous electric field in vivo. This may lead to localized calcium signaling in individual mesenchymal cells, which allow for directional migration. Activities of certain ion channels, electrical coupling amongst cells, and calcium transients may indeed play some crucial role in the differential control of feather bud mesenchymal cells.

Reviewer #1-Question 2: I have a specific questions and remarks below: The vibrating probe experiments seem to show all inward directed currents in the early buds. Electrical circuits need to be closed meaning that the integral of the inward and outward currents must match. Why are no outward current detected?

Reply:

*Thank you for this insightful comment. Indeed the electrical circuits are complete in an organism where currents in some regions are outward while in other regions are inward. In chicken embryos, outward currents could exist in other skin regions, such as the lateral side or ventral side of the torso. We and others have measured inward and outward currents at different positions in different models, e.g tadpole, *Xenopus* oocytes, eye, and ocular lens. These circuits in most cases of intact animals, are global. For example, in zebrafish embryos, outward currents are detected at the head and front torso, while inward currents exist at the tail and tail-dorsal regions (Reid et al., Nat Protoc. 2007.2(3): 661-69). In intact *Xenopus* tadpoles, small inward currents are detected in all body regions other than the gills, which have large outward currents (Reid et al., Dev Biol. 2009.335(1):198-207). Tail amputation in tadpoles results in large outward currents at the wound site while the current density and direction change at different phases of regeneration (Ferreira et al., Development. 2016: dev-142034).*

Another example is that in very early stage chick embryos, Jaffe and Stern measured inward currents at specific parts of an embryo, and outward currents at other parts of the embryo, completing a current flow circuit (Science. 1979 Nov 2;206(4418):569-71).

We believe the critical events happen when a local electrical current pattern changes. As the reviewer rightly pointed out that focally at stage 31, all currents are inward. Critically, at stage 35, the electrical currents reversed direction to become outward. The manuscript for the first time demonstrates this reversal. One local electric current reversal happens locally before limb bud forms in the Xenopus embryo (Robinson KR. Dev Biol. 1983 May;97(1):203-11). This local circuit formation may thus signify and participate in orchestration of local collective cell behaviors.

In this study, the vibrating probe experiment focused on detecting the electric current endogenous to the dorsal skin regions and discovered the critical reversal of the currents (blue arrow in Fig. 1). Field reversal occurs concurrently with polarized bud elongation; at this time (H&H 35) a polarized focal field is formed, whereas no polarized feather endogenous electric field is formed earlier in development (H&H 31). This change in polarized signaling may stimulate coordinated cell migratory behavior.

We have added part of this reply to the discussion section of the manuscript.

Reviewer #1-Question 3: Are any current fluctuations measured during the calcium transients, i.e. is there any evidence that these calcium oscillations are coupled to large scale membrane depolarisations? It could be instructive to have some measurements of membrane potential using voltage sensitive dyes, although this would be a considerable amount of work.

Reply: Thanks for the suggestion and we did try to visualize membrane potential dynamics in skin explants by voltage sensitive dyes or genetically encoded voltage sensors. Unfortunately the penetration of dyes like DiBAC is very limited in the skin explant. Basically for skins over H&H 31 only the most superficial layer of cells got labelled. We also tried the genetically encoded voltage sensor ASAP1 from Michael Lin's group (St-Pierre et al., 2014). Yet a critical problem is that the dynamic range of these genetically encoded voltage sensors are much smaller than that of the calcium sensors. Therefore visualizing mesenchymal cell membrane potential changes in the skin explant context became extremely challenging. That's the very reason we switched to the vibrating probe assay to detect feather endogenous electric currents.

Reviewer #1-Question 4: Although the authors speak of calcium oscillations there are really only a few

transients. I guess it is possible that there are faster oscillations but they may be difficult to detect since this would require fast imaging.

Reply: The reviewer is absolutely right that the calcium fluctuations could occur at a very wide frequency range from microseconds to minutes or even hours. But because feather buds have a height of over 100 microns and is lengthening over time, it took quite some time for the confocal microscope to scan through the feather bud along the distal - proximal axis. Thus, we were limited to monitor Ca^{2+} dynamics with a temporal resolution of minutes, previously. In the updated manuscript we have adopted a new skin stripe configuration (current Fig. 2a and Supplementary Fig. 8) to improve the temporal resolution for visualizing the Ca^{2+} profile along the feather proximal-distal axis. Briefly, the skin explants were cut into one-bud-wide stripes and mounted on their side in glass-bottom dishes coated with fibronectin and poly-L-lysine. Similar to the previous observation in skin explants, we also see the slow Ca^{2+} transients initiate from posterior-distal part of the feather and propagate in the anterior-proximal direction. The duration and propagation speed of the slow Ca^{2+} transients are quantified and shown in current Supplementary Fig. 8. Besides the slow transients, we also measured the duration of the mosaic, fast Ca^{2+} transients, which have an average duration of about 24 seconds.

Reviewer #1-Question 5: in the experiments with the photoactivatable STIM1 is there a response when the cells are not illuminated, i.e. is it clear that the construct has no detectable activity in the absence of blue light illumination?

Reply: To address the reviewer's concern we have tested the activity of the chicken version of opto-cCRAC in cell culture with a Ca^{2+} sensor excited with red light illumination (current Fig. 4a and Supplementary Video 15). No Ca^{2+} influx was observed in the first 30s without blue light. Notable Ca^{2+} flux occurred after blue light illumination and gradually decreased to baseline levels after the blue light was switched off.

Reviewer #1-Question 6: An important question is whether the calcium signals/depolarisation propagates in a directional manner which would be required for them to give any spatial information. There is a slight hint in the movies that this might be case but a more careful analysis of the spatio-temporal aspects of these transients might show whether this is the case. It would require averaging results from a variety of experiments.

Reply: To better observe the tissue Ca^{2+} dynamics during feather elongation, not only did we do more imaging using the skin explants, we also introduced a new skin strip configuration (current Fig. 2a and Supplementary Fig. 8) to improve the temporal resolution for visualizing the Ca^{2+} profile along the feather proximal-distal axis. Similar to the previous observation in skin explants, we also see the slow Ca^{2+} transients initiate from the posterior-distal part of the feather and propagate in the anterior-

proximal direction. The duration and propagation speed of the slow Ca^{2+} transients are quantified and shown in current Supplementary Fig. 8. Besides for the slow

transients, we also measured the duration of the mosaic, fast Ca^{2+} transients.

On the other hand, we are concerned about a technical issue in which mCherry, an expression normalization factor, often was expressed to higher levels in the anterior region of buds compared to the posterior region. In the skin explant configuration, averages of mCherry intensity levels (20 s in duration, before KCl application) were measured in the anterior and posterior 1/3 of the mesenchymal area, respectively. We cannot exclude a possibility that Ca^{2+} signals could be dampened by excessive Ca^{2+} sensors at the anterior region of buds upon KCl stimulation. Therefore, ROIs were set at whole bud mesenchyme for quantification in the updated manuscript (Fig. 2).

Reviewer #1-Question 7: If possible it would be desirable to correlate the dynamics of the calcium transients with changes in the dynamics and direction of cell migration. This could be done by using the mosaic expression of the calcium indicator as marker for cell tracking.

Reply: It is an intriguing idea to monitor cell movements and calcium dynamics within the same tissue. Yet when we put it in practice we encountered several technical difficulties. The most notable one is that GCaMP6s fluorescence signal is much dimmer than that of H2B-GFP. For the time-lapse movies we took the laser power used for GCaMP6s is 5 times that of H2B-GFP. Therefore if we use the laser power suitable for GCaMP then the H2B-GFP signal will saturate, making cells hard to be segmented. If we use the laser power suitable for H2B-GFP then we can barely see the GCaMP signal. If we use a nuclear-localized fluorophore at another excitation/emission wavelength, then the cellular exposure to laser radiation will increase with the extended imaging time, potentially leading to photo-toxicity.

Reviewer #1-Question 8: it might be interesting/useful for the authors to speculate on how they think that the coupling calcium transients could direct cell migration and how this would account for in the observed bud morphogenesis

Reply: In the updated manuscript we incorporated two hypotheses about the link between synchronized Ca^{2+} oscillations and collective cell migration in the discussion part. 1. Elevated cytoplasmic Ca^{2+} promotes myosin-driven cell protrusions by activating MLCK mediated phosphorylation of the myosin light chain (Markova, O., & Lenne, P. F. Seminars in cell & developmental biology, 2009). 2. Diffusive coupling through gap junctions that synchronizes cell Ca^{2+} profiles works as a relay communication mechanism enhancing sensitivity to gradients of chemotactic cues like FGF (Ellison et al., PNAS, 2016, Song, H. K., Lee, S. H. & Goetinck, P. F. Dev. Dyn., 2004, Lin, C. M. et al. Dev. Biol. 2009).

Reviewer #2 (Remarks to the Author):

The manuscript entitled “Coupling of Biochemical-Bioelectric Signals Mediates Collective Dermal Cell Migration” by Li et al. studies the role of Ca^{2+} dynamics in the collective migration of mesenchymal cells during feather development in the chick. By live imaging and pharmacological and genetic manipulations of signaling molecules the authors find that voltage-gated Ca^{2+} channels and gap junctions are necessary for oriented collective cell migration and appropriate feather formation. Morphogenetic proteins Shh and Wnt influence this signaling by regulating expression of gap junction molecule

Connexin 43. The authors conclude that biochemical and “electrical” signals are intermingled to imprint a spatiotemporal map to the morphogenesis of tissues during development.

The study of the features of Ca²⁺ signaling that are important for tissue morphogenesis is interesting and relevant for many fields including developmental biology, cell biology and cancer. However, I found major issues with the experimental design and the conclusions that the authors extract from the presented results.

Reviewer #2-Question 1: Some of the RNAseq data does not seem to match up with the in situ hybridization data shown in Fig. 1. For instance, Connexin 43, which appears to be relevant for the findings presented in the manuscript, shows an upregulation in mesenchymal cells in RNAseq data when transitioning from stage 31 to stage 35 but the in situ hybridization images do not reflect that change. Also, CACNA1C is only shown in RNAseq for mesenchymal cells and not in epithelial cells, although in situs show robust expression in epithelia and developmental upregulation in epithelia, which doesn't match with the lack of KCl-induced Ca²⁺ response in epithelial cells. These discrepancies are not discussed by the authors.

Reply: *Thanks for the advice. We have replaced the Connexin-43 in situ result for HH stage 35 with a more representative one (current Fig. 1d). One thing worth mentioning is that the RNA-seq result shows an increase of mesenchymal Connexin-43 expression from H&H 31 to H&H35, accompanied by a decrease of epithelial Connexin-43 expression (current Fig. 1c). This decreased epithelial expression is likely what left the reviewer with the impression of data inconsistency. As to CACNA1C, the heat maps in current Fig. 1c only show genes with significantly large changes based on the threshold we chose (fold change > 1.3 and p < 0.05). Because the epithelial CACNA1C expression change did not pass the threshold, it is not shown in the heat map. Additionally, RNA-seq and In Situ hybridization only reflect changes at the transcription level. Although CACNA1C transcripts are detected in the epithelium, it is not equal to the presence of functional VGCC channels. Rather, functional assays like the KCl-induced depolarization is a more reliable piece of evidence for active VGCC channels. Furthermore, the epithelial CACNA1C expression seems to be enriched in the periderm layer, which is a specialized epithelial structure which only transiently exists during embryonic development. It is possible the periderm cells possess different sets of functional channels than other epithelial cells.*

We feel that the protein levels and activity levels are more important measures of channels than evaluations of steady state mRNA level (RNA-seq or in situ hybridization)

Reviewer #2-Question 2: Are the feather tissues innervated? Are there sensory terminals? (Hemming et al., 1994). The expression of channels may come from neural structures. In situ hybridizations (Fig. 1 and Suppl Fig. 1) do not provide the spatial resolution to address that, nor they are a good reflection of the expression of functional channels. Immunohistochemistry and assessment of colocalization with cell identity markers could provide more meaningful information on the specific expression of these channels in different cells.

Reply: We noticed that the nerve fibers were only found surrounding the base of feather buds even at embryonic day 11 (Fig. 24b of Hemming et al., 1994 and Fig. 2 of Cahoon-Metzger et al., 2001). We also stained for nerve fibers using the neurofilament antibody (3A10, DSHB) in H&H31, H&H 34 and H&H 35 feather buds (current Supplementary Fig. 2c). The nerve fibers are located in the mesenchyme underneath the feathers and the amount is very low. Therefore the expression of channels cannot be from neural structures, as many of them are enriched in distal feather mesenchyme. Meanwhile the feather endogenous Ca^{2+} oscillations and the Ca^{2+} response upon KCl stimulation were also observed in the distal feather mesenchyme. The reviewer is right in that immunohistochemistry is a better way to detect the spatial distribution of channels. Unfortunately most channel antibodies were made against human and mouse. For the channel antibodies we tested none have worked in the chicken. That is the very reason why we switched to other alternative approaches such as *in situ* hybridization.

Reviewer #2-Question 3: The response to high KCl seems too slow to come from influx through opening of voltage-gated Ca^{2+} channels. Perhaps neural projections are releasing some signal that in turn induces Ca^{2+} transients in mesenchymal cells?

Reply: We believe the delayed response to KCl is due to the nature of the tissue culture itself. First, the E8-E9 chicken dorsal skin explant is large and thick, usually 3-5 mm wide, 5-7 mm long, and 200-300 μ m in thickness, which is over 1000 times larger than the organoid reported in Ellison et al., 2016 that the reviewer mentioned. Consistent with this, when we cut the skin explants into thinner strips that are mounted on their lateral side (current Fig. 2a-d), the KCl responses were faster and clearly not initiated from where the nerve fibers were. The response in the deep mesenchyme, posterior to H&H 35 feather buds, are muscle precursors (please see Li et al., PNAS, 2013). Another piece of supporting evidence is that the skin response to 2 mM Ca^{2+} after Thapsigargin treatment also occurred after a long-time delay (in the current Supplementary Video 10), which can hardly be explained by neuronal effects (as the CRAC channel is gated by internal cell ER Ca^{2+} levels, not membrane depolarization) but can easily be explained by the tissue property. Last but not least, the physiological Ca^{2+} oscillation initiates from the distal part of the mesenchyme, which lacks neuronal fibers as mentioned previously (current Fig. 5a and Supplementary Fig. 8c).

Reviewer #2-Question 4: It is unclear how the cell cultures compare to explants in terms of cell type composition. It is possible that different Ca^{2+} responses obtained upon addition of KCl are due to a mix of mesenchymal and epithelial cells in culture. The authors should have some reporter of cell identity or stain cells post imaging to understand what type of cell gives which response.

Reply: Mesenchymal and epithelial cells have very notable differences in morphology. To highlight this we did immunostaining for TP63 (epithelial cell marker) and Vimentin (mesenchymal cell marker) in H&H 31-H&H 35 feather buds (current Fig. 1b and Supplementary Fig. 2). The description of epithelial and mesenchymal cell morphology & molecular differences has been added to the figure legends and manuscript. Briefly, epithelial cells are cuboidal shaped and arranged in a honey-comb manner. The cell boundaries are always tightly contacting the neighbors. Mesenchymal cell boundaries are not fully

occupied by the neighbors. They are either bipolar or multi-polar shaped due to development of long protrusions. It is very easy to tell whether there is epithelial cell contamination in the culture as they will adhere together to form sheets. Additionally, we have included the bright field images of cells we did

Ca^{2+} imaging with to highlight their morphology. We also did immunostaining for dissociated single mesenchymal cells after Ca^{2+} measurements and they were Vimentin (mesenchymal cell marker) positive (current Fig. 3e and Supplementary Fig. 7e), while the epithelium derived cell line, HEK293T cells, were Vimentin negative (please see the negative controls below, and this was not included in the current manuscript).

While we tried to classify the subpopulations of the mesenchymal cells with respect to KCl-induced Ca^{2+} responses in dissociated single mesenchymal cells, we discovered that there was a correlation between cell morphology and Ca^{2+} responses. Surface area and aspect ratio measured in VIM staining images suggested

that smaller- and/or thin-shaped cells tended to exhibit KCl-induced Ca^{2+} decreases, while larger- and/or polygonal-shaped cells tended to exhibit KCl-induced Ca^{2+} increases (Supplementary Fig. 7e,f). But this is tentative and not focus of this manuscript. It will require further characterization of these feather mesenchyme and verification.

Reviewer #2-Question 5: The paradigm that the authors used for membrane depolarization consists in adding 100 mM KCl extracellularly. This is not a physiological manipulation. What would be the physiological signal that would depolarize mesenchymal cells?

Reply: We think mechanical force may serve as a physiological cue to depolarize membrane potentials in mesenchymal cells. It has been reported that human gingival fibroblasts exhibit slow oscillations of cytosolic Ca^{2+} levels in response to mechanical stretches (Arora et al., 1994). These Ca^{2+} transients could be completely inhibited by Nifedipine, while KCl stimulation could enhance the amplitude of these transients. Additionally, the expression, subcellular localization and even phosphorylation state of Connexin-43 are sensitive to mechanical stretching (Salameh and Dhein, 2013). Therefore in the updated manuscript we added in discussion about the possibility that cell contraction during migration and anisotropic mechanical stretching through adherens junctions may reshape the topology of the functional gap junction network, and hence the slow Ca^{2+} oscillations can be relayed in a directional manner. Experimentally, we do have some preliminary data that membrane tension induced by low osmotic solution (250 mOsm vs. 300 mOsm) triggers Ca^{2+} transients in both epithelial and mesenchymal cells. Because we are currently developing another manuscript dedicated to mechanical force/membrane tension and feather development, the related experiments will not be shown in this manuscript.

Reviewer #2-Question 6: It seems that the level of expression of mCherry-GCaMP6 is always lower in epithelial compared to mesenchymal cells, based on the intensity of the red and green signal in both cell types, which also results in lower ratio of GCaMP6 fluorescence/mCherry fluorescence. Hence it is unclear whether the different responses and Ca²⁺ activity observed in both cells is due to a technical issue of differences in transfection/expression or distinct Ca²⁺ signaling.

Reply: *When we inject the RCAS virus in embryonic day 3 chicken embryos, most of the virus stayed in the cavity between the dermomyotome and sclerotome, while epithelium itself is too thin to hold a large volume of virus. Hence the mesenchyme usually has a higher chance to be infected than the epithelium. We did our best to look for feather buds with at least some epithelial infections for imaging. To highlight this we added bright field images to the current Fig. 2a and it is easy to tell the recordings indeed have decent epithelial infection. Furthermore, the very purpose of introducing 2A-mCherry and calculating the ratio between GCaMP and mCherry intensity is to normalize the differential virus levels in different tissues. In fact, we have good evidence that differential virus levels are not a limiting factor for reading Ca²⁺ dynamics in tissues. For example, in experiments validating the presence of functional CRAC channels (current Fig. 2i and Supplementary Video 10), although the proportion of cells infected by RCAS virus encoding GCaMP6s-2A-mCherry is higher in mesenchyme than epithelium, both tissues exhibited a clear response to extracellular Ca²⁺ after ER Ca²⁺ store depletion. In contrast, in experiments validating the presence of VGCCs (current Fig. 2a-f, Supplementary Videos 1-5), only mesenchymal cells exhibited Ca²⁺ influx after the administration of KCl. These results were highly reproducible. Additionally, in the 4D Ca²⁺ imaging data (current Fig. 5a, Supplementary Fig. 8c and Supplementary Videos 17-20), both epithelium and mesenchyme were decently infected by the virus. While the slow Ca²⁺ oscillations were only seen in mesenchyme but not epithelium.*

Reviewer #2-Question 7: PMA is not specific enough to conclude on involvement of gap junctions when the drug is utilized in several experiments presented in this manuscript. The results with PMA should be either removed from the paper or left in supplementary material. Moreover, differences in phenotype elicited by PMA and carbenoxolone argue that PMA is hitting on other targets, and it is used at an excessive concentration of 500 nM. It seems that the authors favored the PMA results over the carbenoxolone ones because the former were more dramatic/significant but this is misleading because we are looking at a drug that has a plethora of targets, hence the results become very inconclusive in terms of the role of specific Ca²⁺ signaling molecules.

Reply: *We agree PMA is not as specific as Carbenoxolone or 18- α -GA in inhibiting gap junction mediated intercellular communication, and we specifically pointed out the phenotype differences in the manuscript (Paragraph 2 of the session "Perturbing the physiological Ca²⁺ oscillation patterns in feather elongation alters feather orientation and length"). However PMA has an important feature that Carbenoxolone does not have: inhibition of Connexin-43 expression (Oh et al., 1991). Therefore it is no surprising that PMA was more potent to disrupt feather elongation than Carbenoxolone. We have now moved most of the PMA data to the supplementary material (current Supplementary Fig. 6) as suggested by the reviewer.*

In addition, we include the results of experiments with lentivirus based short-hairpin RNAs to genetically suppress Connexin-43 expression in the updated manuscript (current Fig. 5c). The lenti-shRNA

construct inhibited feather bud elongation in every injected embryo, while in the contralateral side without virus feather elongation was normal (we only injected virus in the left side of the embryo, which is the same as how we injected RCAS virus shown in this manuscript). We also applied another gap junction inhibitor Mefloquine (as suggested in Connors BW, Epilepsy Curr, 2012) to skin explants and it also blocks feather elongation as shown by the alteration of feather aspect ratio (current Supplementary Fig. 9). These new data support the involvement of gap junction mediated cell-cell communications in feather elongation.

Reviewer #2-Question 8: Unfortunately, carbenoxolone is also a “dirty” drug (Connors BW, Epilepsy Curr, 2012) and it doesn’t help that the authors use 150 μ M, which seems excessive. Experiments with genetic inhibition of gap junction/connexin43 function are desirable. There are also more specific inhibitors for gap junctions and connexin 43 that the authors could use.

Reply: *The skin explant is a large-size tissue culture, usually 3-5 mm wide, 5-7 mm long, and 200-300 μ m in thickness. Since the reviewer listed the Ellison et al., 2016 paper, we could simply do a tissue size comparison and carbenoxolone dose comparison to evaluate whether we used an excessive amount. In Ellison et al., 2016, carbenoxolone was applied at 50 μ M on a tissue about 150 μ m wide, 300 μ m long, and only one or two cell layers thick (at best 20 μ m). Thus our tissue is more than one thousand times larger in volume. While the carbenoxolone we applied is only three times that dose. Considering the issue of penetration, consumption, degradation, to us the amount applied is justifiable.*

Furthermore, we used a genetic, lentivirus-based shRNA to inhibit Connexin-43 (current Fig. 5c) and another chemical inhibitor of gap junction communication, Mefloquine (current Supplementary Fig. 9), as suggested in Connors BW, Epilepsy Curr, 2012. In both cases feather elongation was significantly inhibited.

Reviewer #2-Question 9: Figure 2g shows that nifedipine, an L-type voltage-gated Ca²⁺ channel blocker blocks completely the change in [Ca²⁺]_i when KCl is added, suggesting that these channels are responsible for the full response to high KCl. This is not what the authors conclude on. The authors need to revise/clarify this mismatch.

Reply: *Indeed Nifedipine is mainly considered as an L-type VGCC blocker. However it does block T-type VGCCs as well (Shcheglovitov et al., 2015). When we look at the I-V plot (current Supplementary Fig. 5f) we consider it to be more similar to T-type VGCCs. However, we cannot exclude a possibility about a cell population expressing L-type VGCCs due to the limited number of tested cells assayed in the patch clamp experiment. Technically, it was very difficult to patch very flat cells like fibroblasts. Therefore to avoid any conflict here we have revised the manuscript by simply saying the patch-clamp experiments also confirmed the presence of active VGCCs.*

Reviewer #2-Question 10: I am confused about the decrease in [Ca²⁺]_i that the authors report in vitro upon KCl addition in the majority of cells. The authors argue that there is a constitutive CRAC-dependent influx that is sensitive to depolarization. However, CRAC channels are voltage-independent, and although they are believed to be inhibited by intracellular [Ca²⁺]_i increases, the fact that these traces

show no increase in $[Ca^{2+}]_i$ argue for an artifact that results in this drop in GCaMP6 signal rather than any real biological explanation. For instance, a change in focal plane can result in a drop in fluorescence intensity. Moreover, why isn't this response present in explants?

Reply: As CRAC channels are inward rectifiers (I-V plot excerpted from Hoth, M. & Penner, R., *Nature* 355, 1992), their conductance will be reduced upon membrane depolarization if they are already in open states. This is another piece of evidence that mesenchymal cell membrane potential is driven by K^+ channels as 100 mM KCl reliably modulates Ca^{2+} responses. We need to further investigate why our CRAC channels are constitutively/spontaneously active regardless of intracellular Ca^{2+} levels in a future study.

As shown in the left graph, Ca^{2+} responses were simultaneously obtained from four single mesenchymal cells (another data set shown in current Fig. 3g). Therefore, it is hard to say that a change in focal plane can result in a drop in "ratios". Although we did not often see this drop in skin explants, we did observe

biphasic KCl responses in the new skin strip configuration (current Supplementary Fig. 8a, b). Thus we believe the heterogeneity in the mesenchymal response is an endogenous nature of the cells rather than a technical artifact.

Reviewer #2-Question 11: Electrophysiological recordings were done only in cultured cells. It is unclear what is the cellular composition of these cultures. What is the percent of mesenchymal cells? And epithelial? Primary cultures are never 100% pure. Hence, it is unclear how these results correlate with the ex vivo or in vivo model used in this study.

Reply: Using the 2xCMF treatment condition (Jiang et al., 1998) the epithelium and mesenchymal cells can be separated well as the epithelial cells physically attached to each other as a sheet during the separation. Besides, epithelial and mesenchymal cell morphologies are very distinct. Epithelial cells are cuboidal shaped and arranged in a honey-comb manner. The cell boundaries are always tightly contacting the neighbors. Mesenchymal cell boundaries are not fully occupied by the neighbors. They are either bipolar or multi-polar shaped due to development of long protrusions (current Fig. 1b and Supplementary Fig. 2a,b). It is very easy to tell whether there is epithelial cell contamination in the culture as they will adhere together to form sheets. To further address the reviewer's concern we included bright field images of cells used for in vitro measurements (current Fig. 3e-h). We also did immunostaining for dissociated cells used for Ca^{2+} imaging and they were Vimentin (mesenchymal cell marker) positive, while the epithelium derived cell lines like HEK293T were Vimentin negative (current Fig. 3e-h).

Reviewer #2-Question 12: The measurements of direct currents through VGCCs are reported to be from $n=3/8$. What does this mean? 3 out of 8 cells showed these currents? 3 out of 8 showed the reported I-V plot? How does this incidence, determined in vitro, correlate with in vivo studies? What is different about the 3 “successful” cells compared to the other 5?

Reply: *We found that 3 out of 8 successful whole-cell voltage clamp recordings showed the VGCC-like I-V curve. The other 5 did not show any measurable currents. It is extremely difficult to acquire whole-cell voltage clamp recordings from cultured mesenchymal cells due to the flatness of the cells. The success rate was about 10%. Thus 8 successful recording is already pushed to a limit based on the time and resource we have. We also think the differential current response to depolarization is somewhat attributable to the heterogeneity of VGCC expression in mesenchymal cells described in the manuscript.*

Reviewer #2-Question 13: Moreover, the authors state that the I-V plot corresponds to T-type Ca^{2+} channels, however, only L-type VGCC blockers are used throughout the paper. This is confusing and suggest that the in vitro and explant systems that the authors used do not completely agree with each other, questioning the validity/relevance of the in vitro data.

Reply: *Although Nifedipine is widely known as a L-type VGCC blocker, it is also capable of blocking T-type VGCCs at high dose (e.g. it blocks Cav3.2 current with $IC_{50} = 5 \mu m$ in *Xenopus* oocytes. Shcheglovitov et al., 2004). Meanwhile in the RNA-Seq and in situ hybridization data we indeed detected expression of both L-type and T-type VGCCs. To avoid the potential confusion we have revised the manuscript by simply saying the patch-clamp experiment also confirmed the presence of active VGCCs.*

Reviewer #2-Question 14: “Contrary to our expectation that CRAC channels are crucial for mesenchymal cell migration, the most potent CRAC inhibitors failed to block feather elongation (Supplementary Fig. 7). Yet artificial Ca^{2+} oscillations triggered by photoactivatable opto-CRAC channels enhanced feather elongation”. This sentence is confusing but the results themselves are not. Doesn’t the lack of phenotype from treating samples with CRAC inhibitor rule out CRAC channels’ role/importance in feather elongation?

Reply: *We agree with the reviewer that CRAC channels are not necessary players for feather elongation, or there probably exist other channels with redundant functions. Yet CRAC channels could still be one contributor to tissue endogenous bioelectric fields. Meanwhile this doesn’t prevent the opto-cCRAC construct from becoming a useful tool to introduce de novo calcium oscillations and change cell behavior in tissues. For example, in the updated manuscript we showed the experimental results that opto-cCRAC could partially rescue the inhibition of feather polarization and elongation by cyclopamine treatment (current Fig. 7c).*

Reviewer #2-Question 15: In Figure 4c, when half of the embryo was transfected with STIM1, the STIM1-expressing feathers look more like the controls than the “wild-type” feathers in the same animal. Is there a compensatory mechanism? Is it non-cell/feather autonomous? Also, while Fig. 4d,f shows quantitative analysis of feather elongation when STIM1 or GCaMP6 are transfected, this analysis is not provided for drug-treated samples or controls in 4b. Hence, it is difficult to compare different

experimental groups and the significance of the results. The same is the case for Suppl Fig. 7 where quantitative analysis is not provided.

Reply: *We have added the aspect ratio quantification for feather buds with different treatment conditions in the current Supplementary Fig 9. It is worth noticing that the culture time of samples in current Fig. 5b and 4c are different. Fig. 5b samples were cultured in vitro for 4 days (C4), while those in Fig. 4c were only cultured for 2 days (C2). The reviewer's observation is right that the feathers in the region positive for opto-cCRAC on the C2 skin had elongated to a level comparable to feathers on C4 skins. That's exactly what we mean by saying artificial Ca^{2+} oscillations triggered by photoactivatable opto-cCRAC channels enhanced feather elongation.*

Reviewer #2-Question 16: Additionally, is the more rounded feather shape seen in the control of Fig. 4c the same phenotype as for carbenoxolone-treated samples shown in Suppl Fig. 7? Perhaps the phenotype is more a developmental delay rather than a real impairment of feather polarity/morphogenesis?

Reply: *Feather buds on Carbenoxolone-treated samples were not just shorter, many of them also develop abnormal polarities (e.g. pointing laterally instead of posteriorly) similar to that of PMA treated samples, as shown in the whole-mount images in the current Fig. 5b and Supplementary Fig. 9a. The changes of feather aspect ratio are shown in the current Supplementary Fig. 9b. Besides, the differential morphology of carbenoxolone-treated feather buds and wild-type feather buds at an earlier developmental stage could be seen in the sagittal sections. As shown in the current Supplementary Fig. 12c, control feather buds (cultured for 2 days, which is comparable to the control side in the current Fig. 4c) had more prominent width (in anterior-posterior direction) than height (in proximal-distal direction), while feather buds on Carbenoxolone-treated C4 skins were shorter in the anterior-posterior direction.*

Reviewer #2-Question 17: In Fig. 7d a control image of Connexin 43 expression pattern in a wild-type sample is missing. Otherwise, it is not possible to conclude that β -catenin overexpression leads to higher Connexin 43 expression as the authors do.

Reply: *First, when we injected the RCAS virus or lentivirus into chicken embryos, we only inject into the left body side. Thus the contralateral body side served as the best internal control (as the feathers on the two body sides of one individual embryo should be mirror-images in normal conditions). In the previous Fig. 7d (current Fig. 8d) it is clear to see the left body side has dramatically increased Connexin-43 expression than the right (control) side. Second, we did provide an external control in Supplementary Fig. 12a: embryos with RCAS-GCaMP6s injected to the left body side. The Connexin-43 in situ signals were comparable in the left and right body side.*

Reviewer #2-Question 18: There are several instances of repeated results between figures and suppl figures (i.e, Fig. 4 and Suppl Fig. 7; Fig. 5 and Suppl Fig. 8). The authors should avoid redundancy.

Reply: *Thanks for the advice. The redundancy in previous Fig. 4b and Sup Fig. 7 has been resolved in the updated manuscript (current Fig. 5b and Supplementary Fig. 9a). The previous Fig. 5b, c and Sup Fig. 8 (current Fig. 6b,c and Supplementary Fig. 10) were not redundant results. The previous Fig. 5b, c*

characterized the motility patterns of all mesenchymal cells in feather buds, while the previous Supplementary Fig. 8 compares the motility pattern differences between mesenchymal cells in the anterior and posterior feather bud, respectively.

Reviewer #2-Question 19: The authors need to rule out that what they report as changes in cell migration are not a consequence of changes in cell density due to altered proliferation or cell death by performing proliferation and apoptosis assays in control and experimental groups.

Reply: *In our previous studies we have characterized cell proliferation and apoptosis during feather elongation. Both events were rare in feather mesenchyme (Chodankar et al., 2003; Li et al., 2013). To address the reviewer's concerns we did incorporate 2-hr BrdU labeling and TUNEL assays to investigate cell proliferation and apoptosis under different treatment conditions in the updated manuscript, respectively (current Supplementary Fig. 9). The BrdU labeling time was extended to 2 hrs (usually 1 hr or even less for other embryonic development processes) because the proliferation rate is very low in feather mesenchyme as described previously. Compared to the control (1/1000 DMSO in culture media) samples, Carbenoxolone, BTP2 and Cyclopamine treatment did not significantly change the mesenchymal cell proliferation rate. Nifedipine treatment even increased the cell proliferation rate. The gap junction inhibitor Mefloquine (Connors BW, Epilepsy Curr, 2012) repressed cell proliferation. On the other hand, Nifedipine, Carbenoxolone, Mefloquine and BTP2 caused no significant changes in the mesenchymal cell death rate as shown by TUNEL staining. Cyclopamine treatment even reduced the cell apoptosis rate.*

Reviewer #2-Question 20: The authors should define/explain why they call the transients in Suppl Fig. 9 Ca²⁺ spikes unlike in the rest of the manuscript where they call them transients, oscillations. It is quite confusing. Are there different types of transients?

Reply:

The Ca²⁺ spike term came from the Belgacem and Borodinsky 2011 PNAS paper, which inspired us to examine the short-term impact of SHH signaling on the mesenchymal cells. As suggested by the reviewer, we have rephrased "spikes" to "transients".

Reviewer #2-Question 21: Have the authors look at more acute responses to SAG or Shh than 20 min after? Are there changes in the frequency of Ca²⁺ transients with Shh/SAG treatments?

Reply: *It is exactly because we didn't see acute responses to SAG or Shh protein in 1-5 min range that we decided to extend the treatment time to 20 min. Yet we still didn't observe notable changes in Ca²⁺ transient frequency. Additionally, in the current Supplementary Fig. 4b,d,e we demonstrate the short term (3 min) and long term (30 min) administration of either Cyclopamine or SAG did not make notable changes in KCl-induced Ca²⁺ increases.*

Reviewer #2-Question 22: In many experiments the n of samples is equal 2. This is not acceptable and it is mostly in experiments that seem more physiologically relevant in their design. The authors need to

increase the n of these experiments to make a valid conclusion (Figs. 4, 5, 6; Suppl. Figs. 5, 8; Suppl. videos 21-28, 31).

Reply:

For the updated manuscript not only did we do more 4D Ca^{2+} imaging using the skin explants, we also introduced a new skin strip configuration (current Fig. 2a and Supplementary Fig. 8) to improve the temporal resolution for visualizing the Ca^{2+} profile along the feather proximal-distal axis. The current sample number used for calculating the slow Ca^{2+} transient duration and propagation speed is 5.

Reviewer #2-Question 23: In the model/summary Figure 8 the authors state: “Heterogeneously distributed VGCCs and CRAC channels contributed to the inward current observed at posterior-distal part of elongating feathers, Ca^{2+} activated K^+ channels may be elevated by high tissue Ca^{2+} levels and contributed to the outward current at anterior-basal part of feathers”. There is no evidence provided by the authors on either the heterogeneous distribution of VGCCs and CRAC nor the Ca^{2+} -activated K^+ channels or the outward current at anterior-basal part of feathers. Hence, this is over speculative and the authors should stay closer to what their results provide in their model.

Reply: *First, for whole skins, the in situ hybridization data demonstrated heterogeneous RNA expression levels of both VGCC and CRAC channel components in feather mesenchyme (current Fig. 1d). Meanwhile fast and sporadic Ca^{2+} transients could be seen even without addition of KCl in feather bud mesenchyme, which also supports the heterogeneity of tissue endogenous Ca^{2+} channel activities (current Supplementary Fig. 8d). We also tried to elaborate the subpopulations of mesenchymal cells based on intrinsic properties of both channels in response to membrane depolarization. As shown in current Fig. 3g,h, it is very difficult to conclude whether there is a cell population expressing either VGCC or CRAC channels in regular Ringer’s solution as baseline Ca^{2+} fluctuations correlate with KCl-induced Ca^{2+} responses. Therefore, we used Thapsigargin with 0 mM Ca^{2+} solution and indeed the condition flattened out the baseline Ca^{2+} levels (R_0 of Fura-2 is about 0.5 in all cells measured in current Supplementary Fig. 7a,b). Ca^{2+} stimulation after Ca^{2+} deprivation were observed in the dissociated mesenchymal cells, confirming the non-uniform presence of CRAC channels*

Second, our assumption is that all CRAC channels should be activated by the ER depletion. If a cell expresses only CRAC channels, no response to KCl should be observed. As for a cell expressing only VGCC, no response to 2 mM Ca^{2+} alone should be observed. As expected, we observed four different groups: 1) no response under both conditions 2) only 2 mM Ca^{2+} alone-induced Ca^{2+} increases 3) only KCl-induced Ca^{2+} increases 4) responses to both conditions. We also have a concern that 100 mM KCl may not exclusively distinguish cell populations as we cannot be sure that membrane potential changes to “completely” block all CRAC channels (see the I-V plot in question 10). We are very cautious to interpret our data due to its complexity.

Third, we discovered that the heterogeneous cell populations can be further identified by different cell morphologies in different parts of a feather bud (current Supplementary Fig. 2). More bipolar shaped cells were observed in the anterior bud mesenchyme while more multipolar cells were

observed in the posterior bud mesenchyme. What's more interesting is that the cell morphologies correlate with KCl-induced Ca^{2+} responses in vitro (current Supplementary Fig. 7e,f). Taken together, there is evidence from multiple aspects that support our model. We believe the variations in the cell subpopulations contributed to coordinated tissue morphogenesis.

As to Ca^{2+} -activated K^+ channels, the expression of KCNMA1 has been detected in the anterior-basal part of H&H 35 feather bud but not at earlier stages (current Fig. 1d). Meanwhile the outward current has been detected by a vibrating probe in the anterior feather bud at H&H 35 but not at earlier stages (current Fig. 1a).

Reviewer #2-Question 24: The authors claim in the opening paragraph of the Discussion that “To our knowledge, the discoveries here are the first report of slow, multicellular synchronized Ca^{2+} oscillations to coordinate the collective cell movement patterns in tissue morphogenesis”, but in fact many reports have demonstrated this in different contexts. To name a few, it has been shown that multicellular Ca^{2+} transients are present during neurulation and are necessary for neural tube morphogenesis (Suzuki et al., Development 2017) and that bidirectional radial Ca^{2+} activity regulates neurogenesis and migration during early cortical column formation (Rash et al. Science Advances 2016). The authors should cite previous studies (others are: Ellison et al, PNAS 2015; review: Markova and Lenne, Seminars in Cell and Developmental Biology 2012) and revise their discussion. The originality of this study is somewhat overstated.

Reply: *The studies listed by the reviewer are very helpful. We have incorporated them in different parts of the discussion. For example, we rephrased the statement highlighted by the reviewer (currently in the second to the last paragraph of the discussion) as below: “ Ca^{2+} fluctuations have previously been implicated in modulating cell migration¹⁻³, convergent extension⁴, apical constriction⁵, etc. Our study demonstrates slow multicellular Ca^{2+} oscillations coordinate collective mesenchymal cell migration in skin appendage organogenesis.” The references have been updated accordingly as well. Currently Ref 55 is Markova & Lenne 2012; Ref 54 is Rash et al., 2016; Ref 56 is Suzuki et al., 2017; Ref 58 is Ellison et al., 2015.*

Reviewer #2-Question 25: In page 15, 3rd sentence from top should refer to Supplementary Fig. 10b, not “11b”.

Reply: *We have revised the manuscript accordingly, in current version it is Supplementary Fig. 12b.*

Reviewer #2-Question 26: In page 15, 7th sentence from bottom should refer to Supplementary Fig. 10c, not “10b”.

Reply: *We have revised the manuscript accordingly. In current version it is Supplementary Fig. 12c.*

Reviewer #3 (Remarks to the Author):

In this manuscript Li et al. test the hypothesis that calcium signaling dynamics mediate coordinated

mesenchymal cell migration using the developing chick feather bud as their main experimental model. It is well known that calcium signaling is a critical mediator of multiple developmental processes however specific mechanisms for how calcium signals are propagated and their functional significance have been elusive, primarily due to the extreme dynamic nature of calcium and ion currents in live cells and tissues. In this study the authors use a robust and previously tested explant model system and a range of powerful molecular and optogenetic approaches coupled with multi-dimensional live imaging to study the role of calcium oscillations during feather development. Overall, the novelty, breadth and stringency of the experimental approaches and the broad biological implications of the findings make this manuscript appropriate and valuable for the general readership of Nature Communications.

As a general comment I find the feather explant model the authors have previously developed and validated to be exceptionally useful for its biological relevance, genetic amenability and suitability for live imaging approaches. The quality of the live imaging data throughout the study is very good and the various reporters used to visualize calcium and other cell activities have been previously validated and appropriate.

Reviewer #3-Question 1: However, the authors refer to the epithelium and mesenchyme of the feather bud throughout the manuscript without clearly defining the two populations molecularly or even morphologically. This is problematic when evaluating the live imaging data, in the absence of secondary markers. Although it is intuitive to distinguish the two populations by location within the tissue structure or by referring to the literature it would be useful that the authors more clearly define them, perhaps in a Fig1 subpanel to assist the non-expert reader and remove any ambiguity. This is especially important for the mesenchyme that by nature is thought to be a heterogeneous population in multiple tissues.

Reply: *Thanks for the advice. To highlight the morphological and molecular differences of epithelial and mesenchymal cells, we did immunostaining for TP63 (epithelial cell marker) and Vimentin (mesenchymal cell marker) in H&H 31-H&H 35 feather buds and imaged samples at different resolutions to highlight morphological differences of these cells (current Fig. 1b and Supplementary Fig. 2). The description of epithelial and mesenchymal cell morphology & molecular differences has been added to the figure legends and manuscript. Briefly, epithelial cells are cuboidal shaped and arranged in a honey-comb manner. The cell boundaries are always tightly contacting the neighbors. Mesenchymal cell boundaries are not fully occupied by the neighbors. They usually develop long filopodia. Anterior mesenchyme has lots of elongated, bipolar cells aligned along the epithelial-mesenchymal boundary. Posterior mesenchymal cells are mainly multipolar. We also added bright field images of feather buds used for skin strip assays in the current Fig. 2a to highlight the configuration of feather epithelium and mesenchyme.*

Reviewer #3-Question 2: The data in Figure 4 are convincing and clearly define the significance of this study in defining a specific role of calcium signaling in feather bud development. Subsequent figures provide some possible insight on the underlying molecular mechanism that regulates calcium currents and role of Hh signaling. However, in these experiments (figure 6) it is unclear if asynchronous calcium oscillations are the primary effect of the impaired SHH or the secondary effect of other cell interactions in the explant. Subsequent studies are needed with cell type specific knockdown of SHH signaling and

rescue to show that this is affecting movement of mesenchymal cells directly but these are clearly beyond the scope of this manuscript.

Reply: *To address the reviewer's concern we treated the opto-cCRAC infected skin explants with Cyclopamine and cyclic blue-light illumination. Then we observed the feather bud morphology in areas infected by the virus to different degrees after 2 days (current Fig. 7c). In those areas without virus infection or with only sparse infection, the Cyclopamine induced inhibition of feather polarization and elongation was very notable, while those areas with enriched mesenchymal infection became new feather tips, which significantly increased the feather aspect ratio. Interestingly, the rescue phenomenon could occur even when the highly infected area only marginally overlaps with a feather bud. Therefore although we were not able to do cell-type specific knockdown and rescue experiments at this point, this partial rescue experiment indeed supports the involvement of Ca^{2+} signaling in the regulation of feather elongation by SHH signaling.*

Reviewer #3-Question 3: In the culture experiments of figure 3 it is not really clear how these cells were cultured and while I believe the claims about the data, these results could easily be artefactual. They could be a result of dysregulation resulting from dissociation or a number of other reasons. Moreover, the mesenchymal population itself is very likely heterogeneous, not simply their response to KCl. If I am reading this correctly, the contribution of this to the paper as a whole is that mesenchymal cells at this stage are a heterogeneous population, implying that because they respond differently to depolarization their connectivity is essential to their functioning as a group to mobilize in a directed pattern. The authors may want to use the RNA-seq data to determine whether there are subpopulations of mesenchymal cells here and mention that they are marked by different expression of these calcium related genes. However, this may also be beyond the scope of this paper but a more thorough interpretation of the data and the caveats relating to fig 3 may need to be discussed in the manuscript.

Reply: *Thanks for the suggestion. To fully characterize the differential subpopulations of mesenchymal cells, single-cell RNA-Seq is required. Yet the transcriptomic comparison may not reflect the whole story as the expression of a channel doesn't guarantee it to be functional (e.g. we detected voltage-gated calcium channel expression in epithelial tissue by bulk RNA-Seq but tests with chemicals indicated functional VGCCs were located in the mesenchyme).*

To address the reviewer's concern we included several new experiments in the updated manuscript to demonstrate the heterogeneity of mesenchymal cells based on morphology and channel activities: 1. We did Vimentin (mesenchymal cell marker) staining and observed mesenchymal cells in different parts of the feather bud exhibit different morphologies (current Supplementary Fig. 2a,b). More bipolar shaped cells were observed in the anterior bud mesenchyme while more multipolar cells were observed in the posterior bud mesenchyme. 2. We introduced a new skin strip configuration (current Fig. 2a and Supplementary Fig. 8) to improve the temporal resolution for visualizing the Ca^{2+} profile along the feather proximal-distal axis. Briefly, the skin explants were cut into one-bud-wide strips and mounted on their side in glass-bottom dishes coated with fibronectin and poly-L-lysine. Similar to the previous observation in skin explants, we also see the slow Ca^{2+} transients initiate from the posterior-distal part of the feather and propagate in the anterior-proximal direction. When we perfuse KCl to the skin using this configuration, it is clear that the increase of cytosolic Ca^{2+} is not isotropic in feather mesenchyme. Some cells even exhibited biphasic cytosolic Ca^{2+} level changes like what has been observed in dissociated cell culture (current Fig. 3a,b and Supplementary Fig. 8a,b). Furthermore, we did more time-lapse ratiometric

Ca²⁺ imaging using the skin strips without any chemical blocker treatment at different temporal resolutions. Not only did we observe the slow Ca²⁺ transients previously seen using the explant setup, we also saw fast, sporadic Ca²⁺ transients in feather mesenchyme (current Supplementary Fig. 8d). These new experimental results support the idea that the heterogeneity of channel activities observed in the dissociated cell culture also exists in the tissue context.

Reviewer #3-Question 4: Minor comment: Reference 16 in the text appears to be citing the wrong paper.

Reply: *We have revised the manuscript. In the current version it is Reference 11: Reid, B., Nuccitelli, R. & Zhao, M. Non-invasive measurement of bioelectric currents with a vibrating probe. Nat. Protoc. 2, 661-669 (2007).*

Overall the data presented in this manuscript significantly advance the field and our understanding of the role of ion currents in regulating developmental growth and patterning processes and may serve as an important stepping stone for further in vivo studies to fully resolve the underlying genetic and molecular mechanisms. As such I recommend that this manuscript is accepted with minor changes based on my comments above.

Reviewers' Comments:

Reviewer #1:

Remarks to the Author:

The revised version of the manuscript has tried to address some of the critique from the various referees.

The response to most of my questions has remained unfortunately somewhat hand waiving (questions,1,2,3) or due to difficulties in obtaining further data.

The authors have made an attempt to improve the characterisation of the spatio-temporal dynamics of the calcium transients via introducing a slice culture that allows faster imaging. This may help in future investigations. It has however so far not been able to address some specific questions, such as for example my question 6.

Question 6, asks whether there is evidence that the calcium oscillations/depolarisation propagate in a wave like fashion that could be able to direct the movement of the cells. I would expect to see a graph of amplitude versus space (posterior to anterior) with a number of activity profiles taken at various times, showing a moving wave front or something similar. I cannot find data that address this in the answer in the rebuttal letter (I do not understand what the graph is supposed to show, it shows a correlation of some kind) and not see relevant information in sup fig 8. What does sup fig 8e show, what are the red bar and the different populations of dots? The legend of this critical figure should be improved. Answering this question as some of the others clearly needs more work

However despite this I still think this is an interesting innovative piece of work that should be published.

Reviewer #2:

Remarks to the Author:

The manuscript entitled "Orienting Feather Buds by Novel Mechanisms Coupling Biochemical-Bioelectric Signals" by Li et al. studies the role of Ca^{2+} dynamics in the collective migration of mesenchymal cells during feather development in the chick. By live imaging and pharmacological and genetic manipulations of signaling molecules the authors find that voltage-gated Ca^{2+} channels and gap junctions are necessary for oriented collective cell migration and appropriate feather formation. Morphogenetic proteins Shh and Wnt influence this signaling by regulating expression of gap junction molecule Connexin 43. The authors conclude that biochemical and electrical signals are intermingled to imprint a spatiotemporal map to the morphogenesis of tissues during development.

The study of the features of Ca^{2+} signaling that are important for tissue morphogenesis is interesting and relevant for many fields including developmental biology, cell biology and cancer. The authors addressed all the concerns raised on the original submission by performing additional experiments and revising the text. The findings presented in the revised manuscript are convincing and make a significant contribution to the field of calcium signaling-mediated morphogenesis.

Reviewer #3:

Remarks to the Author:

The revised manuscript is greatly improved and effectively addresses all my main concerns and suggestions. I find the immunohistochemistry experiments very useful to distinguish and further characterize the epithelial and mesenchymal cell types involved in these experiments. The

Cyclopamine experiments are sufficient at this point and provide further confidence for the interpretation of the data. Perhaps in a follow-up study the authors can dissect the specific mechanisms using in vivo genetic approaches. The authors also provide additional experimental data to characterize the heterogeneity within the mesenchymal population in the tissue and convincingly demonstrate consistency between the in vitro and explant results. My general impression is that the authors went above and beyond to address all the reviewers comments and as a result the revised manuscript provides better clarity. Despite any possible shortcomings I strongly believe that this study is important and should be accepted for publication in its current form.

Reviewer #1 (Remarks to the Author):

The revised version of the manuscript has tried to address some of the critique from the various referees.

The response to most of my questions has remained has remained unfortunately somewhat hand waiving (questions,1,2,3) or due to difficulties in obtaining further data.

The authors have made an attempt to improve the characterisation of the spatio-temporal dynamics of the calcium transients via introducing a slice culture that allows faster imaging. This may help in future investigations. It has however so far not been able to address some specific questions, such as for example my question 6.

Question 6, asks whether there is evidence that the calcium oscillations/depolarisation propagate in a wave like fashion that could be able to direct the movement of the cells. I would expect to see a graph of amplitude versus space (posterior to anterior) with a number of activity profiles taken at various times, showing a moving wave front or something similar. I cannot find data that address this in the answer in the rebuttal letter (I do not understand what the graph is supposed to show, it shows a correlation of some kind) and not see relevant information in sup fig 8. What does sup fig 8e show, what are the red bar and the different populations of dots? The legend of this critical figure should be improved. Answering this question as some of the others clearly needs more work

However despite this I still think this is an interesting innovative piece of work that should be published.

Reply: Indeed, Question 6 from Reviewer #1 is crucial for our future investigation on the deeper mechanism of how tissue-wide calcium dynamics convey directionality information. The authors of this manuscript have also gathered to discuss about this several times. However, no consistent opinion has been achieved in describing the phenomenon as Ca^{2+} waves, mainly because: 1. Some cells at posterior-distal mesenchyme maintained high cytosolic Ca^{2+} throughout the time span we observed, making this phenomenon different from the traveling ATP waves seen in the collectively migrating Dictyostelium. 2. when we plotted the GCaMP6s/mCherry ratio profile in anterior and posterior mesenchyme separately (Fig. 5a), there is barely any delay in the peaking time of cytosolic Ca^{2+} in the anterior mesenchyme compared to that of the posterior mesenchyme, implying the phase of Ca^{2+} oscillation is generally "synchronized" between the mesenchymal cells linked by the gap junction network. In this updated version we also compared the temporal changes of Ca^{2+} profile in anterior vs posterior bud mesenchyme under skin strip configuration (current Supplementary Fig. 8d). The result is consistent with previous discovery. Yet it could be possible that the Ca^{2+} propagation occurs so fast that it cannot be efficiently captured by the time-lapse imaging (with time interval at minute scale), therefore we are considering applying dual-channel light sheet confocal microscopy (higher spatiotemporal resolution) to study this phenomenon in the future. 3. We did observe a gradual expansion of the high Ca^{2+} zone in feather

mesenchyme during feather elongation (Fig. 5a). When we quantify the speed of the expansion (Supplementary Fig. 8f), it is slower than the spreading speed of Ca²⁺ waves reported previously (usually several μm per second, see Markova & Lenne, 2012 semcdb), thus this speed more likely represent the pace of gap junction network establishment rather than the diffusion rate of Ca²⁺ ion. In the updated version we added this statement to discussion. In sum we are still looking for more solid evidence to support this phenomenon as Ca²⁺ wave propagation.

The red bar and the dots are just a customized boxplot to fit the publication request of the journal. This is described in the legend of Supplementary Fig. 8b: “Customized boxplot: Mean (red) ± SD (pink), 95% confidence interval (violet). Dots denote individual data points.” In the updated version we also increased the dot size in Supplementary Fig. 8f to make them easier to see.

Reviewer #2 (Remarks to the Author):

The manuscript entitled “Orienting Feather Buds by Novel Mechanisms Coupling Biochemical-Bioelectric Signals” by Li et al. studies the role of Ca²⁺ dynamics in the collective migration of mesenchymal cells during feather development in the chick. By live imaging and pharmacological and genetic manipulations of signaling molecules the authors find that voltage-gated Ca²⁺ channels and gap junctions are necessary for oriented collective cell migration and appropriate feather formation. Morphogenetic proteins Shh and Wnt influence this signaling by regulating expression of gap junction molecule Connexin 43. The authors conclude that biochemical and electrical signals are intermingled to imprint a spatiotemporal map to the morphogenesis of tissues during development.

The study of the features of Ca²⁺ signaling that are important for tissue morphogenesis is interesting and relevant for many fields including developmental biology, cell biology and cancer.

The authors addressed all the concerns raised on the original submission by performing additional experiments and revising the text. The findings presented in the revised manuscript are convincing and make a significant contribution to the field of calcium signaling-mediated morphogenesis.

Reply: We thank Reviewer #2 for the positive feedback.

Reviewer #3 (Remarks to the Author):

The revised manuscript is greatly improved and effectively addresses all my main concerns and suggestions. I find the immunohistochemistry experiments very useful to distinguish and further characterize the epithelial and mesenchymal cell types involved in these experiments. The Cyclopamine experiments are sufficient at this point and provide further confidence for the interpretation of the data. Perhaps in a follow-up study the authors can dissect the specific mechanisms using in vivo genetic approaches. The authors also provide additional experimental data to characterize the heterogeneity within the mesenchymal population in the tissue and convincingly demonstrate consistency between

the in vitro and explant results. My general impression is that the authors went above and beyond to address all the reviewers comments and as a result the revised manuscript provides better clarity. Despite any possible shortcomings I strongly believe that this study is important and should be accepted for publication in its current form.

Reply: We thank Reviewer #3 for the positive feedback. Yes the in vivo genetic approaches will be included in the follow-up studies.